# AquaX: An enhanced and revised AquaMaps framework to model marine species distributions and biodiversity

Gabriel Reygondeau[1]☯*, Yulia Egorova[1]☯*, Kristina Boerder[2], Derek P. Tittensor[2], Kristin Kaschner[3], Kathleen Kesner-Reyes[4], Nicolas Bailly[4,5], William W. L. Cheung[5]

1 Rosenstiel School of Marine, Atmospheric, and Earth Science, University of Miami, Miami, Florida, United States of America, 2 Future of Marine Ecosystems Lab, Dalhousie University, Halifax, Canada, 3 Department of Biometry and Environmental Systems Analysis, Albert-Ludwigs University, Freiburg im Breisgau, Germany, 4 Quantitative Aquatics, Inc., Los Baños, Laguna, Philippines, 5 Institute for the Oceans and Fisheries, University of British Columbia, Vancouver, British Columbia, Canada

☯ These authors contributed equally to this work.
* gabriel.reygondeau@gmail.com (GR); egorova.uu@gmail.com (YE)

## Abstract

Marine biodiversity underpins ecosystem health and is critical for the provision of essential ecological services. Global efforts to mitigate biodiversity loss are underway but require comprehensive knowledge on the biogeography of species to be effective. However, key challenges limit comprehensive mapping of species distributions, including the ecosystem complexity and difficulty of sampling the marine realm. Global initiatives such as AquaMaps pioneered large-scale marine species mapping using species distribution models or ecological niche models and provided the knowledge base for effective marine conservation and management. Recently, methodological and data advances have enabled a more modern and robust approach that enables higher resolution outputs more suited to conservation applications at all scales. Building on AquaMaps, we developed a next-generation marine species habitat suitability modelling platform called AquaX, providing a suite of advances that include an ensemble of ten machine learning algorithms, enabling spatial uncertainty assessments, validation indices, and ecological niche representation at a ten-fold improved spatial resolution of 0.05°. Furthermore, AquaX integrates (i) accepted taxonomy from the World Register of Marine Species, (ii) species-specific ecological, physiological, and biogeographical information (D3-Ocean system), (iii) updated occurrence records validated through expert input, and (iv) refined species range maps using expert knowledge and biogeographical divisions. AquaX also projects species' habitat suitability for both present and future conditions based on two time periods and three climate scenarios. This work provides species range maps for numerous species compared to previously available datasets and improves the accurate use of observational data. The approaches described here improve predictive accuracy at scales more relevant to marine biodiversity conservation and offer

**Data availability statement:** All data underlying the results presented in this study and Supporting information are openly available on Figshare at https://doi.org/10.6084/m9.figshare.30754358. The AquaX framework scripts used to generate the ensemble habitat suitability projections can be requested directly through the AquaMaps website (https://www.aquamaps.org) in accordance with their data-use and code-access policies.

**Funding:** The author(s) received no specific funding for this work.

**Competing interests:** The authors have declared that no competing interests exist.

an openly accessible tool to support marine biodiversity research and conservation planning under accelerating environmental change. AquaX represents an important step forward in species distribution modeling, enabling researchers and policymakers to better understand marine biodiversity patterns and develop more effective conservation strategies.

## Introduction

Biological diversity, the variety of forms of life from genes to ecosystems, is considered one of the most important dimensions to assess the state and health of our environment, both terrestrial (United Nations Sustainable Development Goal 15; SDG15) and aquatic (SDG14), and track our progress towards global sustainable development [1]. Since the Industrial revolution, anthropogenic pressures have drastically altered marine ecosystems, notably through overexploitation, climate change, pollution, habitat destruction and invasive species, leaving only about 13% of the ocean in largely untouched condition and placing one in every eight evaluated species at risk of extinction [2,3]. Biodiversity loss is closely linked to impacts on ecosystem services and benefits such as food provisioning and carbon sequestration. Global pledges have been made to "halt and reverse nature loss by 2030" through the Kunming-Montreal Global Biodiversity Framework [4]. Effective implementations of such efforts require indicators for biodiversity used to monitor biodiversity loss and recovery and detect ecological changes across space and time. Examples of widely-used biodiversity indicators include species richness, species turnover, and extinction risk [5]. Generally, to 'bend the curve' of biodiversity loss [3], it is critical to improve and deepen our understanding of the composition and distribution of marine biodiversity to be able to determine effective conservation priorities and pathways toward sustainability.

The complexity of marine ecosystems and the operational, logistic, and financial challenges of conducting oceanic surveys have historically limited our ability to fully comprehend global ocean biodiversity dynamics [6]. However, the emergence of freely accessible international data platforms such as the Ocean Biodiversity Information System (OBIS), Global Biodiversity Information Facility (GBIF), which provide species occurrence data at high spatial and temporal resolutions, combined with advances in statistical tools and modelling approaches, have opened new frontiers in marine biogeography and conservation research.

In this context, species distribution models (SDMs), which relate species occurrences to environmental conditions, have become a critical tool, enabling researchers to project species distributions and explore patterns of biodiversity [7]. SDMs estimate the relative suitability of environmental conditions for a species based on its ecological niche, which can be interpreted as an index of habitat suitability and threshold to predict potential presence or absence [7,8]. These models are particularly valuable to project into areas where direct observations are sparse, costly or hard to obtain. By projecting species distributions and identifying ecological niches,

SDMs can inform conservation planning, fisheries management, environmental policy, and through future projections, climate change risk assessments.

SDMs have become central to marine biogeography because they allow inference of species–environment relationships and prediction of distributions where observations are sparse. A systematic review of 236 marine SDM studies showed that SDMs are now widely applied but that robust practice requires transparent data handling, consistent evaluation metrics, and explicit reporting of uncertainty [9]. Recent syntheses highlight major methodological advances, including increased use of machine-learning algorithms, three-dimensional and depth-explicit modelling, and improved handling of dynamic oceanographic variability, while also noting persistent biases in occurrence data and challenges linked to the spatial–temporal dynamics of the ocean [10]. When SDMs are projected under future climate scenarios, structural model differences and extrapolation into novel environmental conditions contribute substantial uncertainty, which can exceed Earth system model uncertainty at multi-decadal to century scales, underscoring the need to quantify uncertainty when such projections are used for management or climate-risk assessments [11]. Together, these developments establish the current expectations for marine SDMs and provide the context in which next-generation frameworks such as AquaX have been developed.

AquaMaps (https://www.aquamaps.org/) pioneered the global application of species distribution models across large numbers of marine species by developing a simple yet robust model that attempts to capture the ecological niche concept introduced by Hutchinson [12]. This approach uses an environmental envelope, based on trapezoidal response curves for multiple environmental factors that represent species' habitat preferences. Each response curve assumes that the species-specific habitat suitability is highest (scaled to 1.00) within its preferred environmental range and decreases linearly toward zero at the thresholds where conditions become unsuitable for survival. This has allowed AquaMaps to produce large-scale global habitat suitability maps for thousands of marine species, providing an essential and open resource for biodiversity research and ecosystem management since 2010.

Despite its widespread use, AquaMaps has faced criticism regarding the simplification of species-environment relationships to simple trapezoids, reliance on coarse environmental data (0.5° x 0.5° grid cells), and a lack of validation metrics [13]. These limitations can lead to inaccuracies in species distribution estimates, particularly for species with complex ecological requirements or those with few recorded observations. As our understanding of marine ecosystems improves and more detailed biological and environmental data become available, updating and expanding on the original methodologies of AquaMaps makes it possible to enhance predictive accuracy and better capture the nuances of species-environment interactions. Incorporating more sophisticated modelling techniques and finer-scale data would also strengthen the reliability and applicability of AquaMaps in marine biodiversity research and makes it more relevant to the scales at which conservation planning typically occurs.

AquaX is a next-generation framework for marine species distribution modelling that builds on two decades of work by AquaMaps, advancement of SDM algorithms, and availability of environmental data. AquaX improves key components of the original AquaMaps model by incorporating recent advances in spatial statistics and machine learning including: (i) employing an ensemble of ten distinct algorithms to predict species' distributions, enabling model quality assessments, the computation of spatial uncertainty, and a detailed representation of ecological niches; (ii) providing projections of future species distributions for multiple time periods and climate change scenarios, specifically mid-century and end-century under three Shared Socioeconomic Pathway (SSP) scenarios; and (iii) projecting current and future distributions at a 5 km resolution, allowing for the representation of local and regional features highly sought after by conservation scientists.

AquaX brings a modular and transparent framework for curating and integrating species-level information. This framework enables AquaX to (i) use the accepted taxonomy for each species following the World Register of Marine Species (WoRMS, www.marinespecies.org); (ii) compile all the new updates on ecological, physiological, and biogeographical information for each species, including habitat preference, depth range, and general distribution; (iii) verify occurrence

records for each species from all available databases, evaluating the reliability of each record based on ecological evidence, expert-provided range maps, or expert opinion; and (iv) constrain or refine species' habitat suitability maps based on available expert knowledge and by integrating gathered information with biogeographical divisions tailored to each species' habitat.

This paper aims to present the AquaX framework, discussing how it marks a significant step forward in marine species distribution modeling, providing a more robust, openly accessible and flexible framework to support biodiversity research and conservation planning in the face of environmental change. We discuss the application of AquaX in three areas of ecological and conservation studies: (1) expert spatial delineation of species ranges and refinement of biodiversity distributions to support conservation planning; (2) high-resolution mapping of current and future species habitats to assess climate-driven shifts, local extinctions, and invasions; and (3) global-scale biodiversity assessment to enhance macroecological analyses and inform marine management and policy. We also examine the key limitations of the framework and opportunities for future development.

## Materials and methods

### Occurrence data

To enable consistent and reliable retrieval of occurrence data across multiple sources, we first standardized species names prior to data compilation. Each species was identified using its Aphia unique identifier (AphiaID), a unique and persistent identifier assigned by the World Register of Marine Species [14] (WoRMS), which provides a standardized scientific name and taxonomic classification. The dataset of all accepted species names and corresponding AphiaIDs was obtained from the WoRMS team and downloaded in September 2024.

AquaX updated species occurrence data with multiple sources of publicly available databases. Occurrence data were compiled from three widely used biological databases: AquaMaps [15], the OBIS [16], and the GBIF [17]. Full process is shown in S1 Fig. AquaMaps data, provided directly by the AquaMaps team (October 2024), were pre-processed to generate individual species-level occurrence files. Each record was standardized taxonomically using the WoRMS taxonomy via the Aphia unique identifiers (AphiaID). OBIS records were retrieved using the *robis* R package [18], and GBIF data were obtained using the *rgbif* package [19,20]. For AquaMaps and OBIS, records were queried using the species' unique AphiaID, while GBIF data were accessed using scientific names due to the absence of AphiaID integration. For all datasets, we extracted relevant fields, including the species identifier (AphiaID), geographic coordinates (latitude and longitude), temporal details (month and year of collection), record type (e.g., observation, specimen), and available quality flags indicating record reliability.

AquaX integrated important features of both AquaMaps and OBIS datasets in the inclusion of data quality indicators. To standardize quality indicators across data sources, we implemented a unified binary flagging system based on the AquaMaps convention (S1 Fig). Such indicators help distinguish between verified and potentially erroneous records, thereby facilitating the assessment of data reliability. Specifically, record quality is flagged using a binary system, with values of 1 indicating verified occurrences and 0 denoting erroneous entries. For AquaMaps records, the existing binary flags were directly adopted. OBIS employs a more complex quality control scheme, wherein each occurrence record is associated with a comma-separated list of quality flags indicating potential issues (e.g., invalid coordinates, depth anomalies, terrestrial locations). Occurrence records flagged in the OBIS data as located on land, missing geographic coordinates, or placed in physically implausible locations were assigned a value of 0. Other original OBIS flags, such as missing depth information or occurrences falling outside known bathymetric limits, were retained in original flag column but not automatically considered erroneous, and their value was left blank in standardized binary flag column. In contrast, GBIF does not include a dedicated quality flag column within the downloaded dataset. However, during data retrieval, only occurrence records with valid geographic coordinates (latitude and longitude) and no reported spatial issues were included, using the GBIF download filters to ensure baseline data reliability.

Since GBIF does not include explicit quality flags in its dataset, and only records without spatial issues and with valid latitude and longitude were downloaded, the flag column for all GBIF entries was left blank to reflect the absence of a source-provided quality classification.

After combining data from AquaMaps, OBIS and GBIF, geographic coordinates were rounded to the 4th decimal point and points with problematic geospatial location (0,0 coordinates) were removed. Then, duplicate or triplicate entries were identified by cross-referencing across the datasets. In some instances, records with identical coordinates and collection dates carried quality flags in the AquaMaps or OBIS datasets but remained unflagged in GBIF. To ensure consistency and avoid conflicting information, data were grouped based on key attributes, including AphiaID, geographic coordinates, and collection date. This grouping allowed for the identification of duplicated entries and retention of the most complete entry (i.e., entries with quality flag and basis of record).

**Ecological and biogeographical information**

In addition to WoRMS taxonomic data, ecological, biological and biogeographic information (e.g., depth range and habitat type) were gathered from FishBase [21] (FB) for fishes or SeaLifeBase [22] (SLB) for other marine organisms using the R package *rfishbase* [23]. Then, information on the availability of IUCN expert range maps [24] (downloaded September 2024) and total unique number of occurrence points were also added.

To precisely describe the habitat of a given species in three dimensions, we introduced the Distance-3 Ocean System (D3OS) inspired by the FB and SLB habitat categories (Fig 1). This system incorporates three key spatial descriptors (classification diagram is shown S2 Fig):

- *Distance to bottom* defines the affinity of a species to the water column (pelagos) or the seafloor (benthos). This classification uses habitat information obtained from FB or SLB and literature. Species were categorized into three groups: pelagic, demersal and benthic. The pelagic category includes species that inhabit the open water column without direct interaction with the seabed, including those described as pelagic, pelagic-oceanic, bathypelagic, or pelagic-neritic in FB and SLB. The demersal category comprises species that live near or just above the seabed but do not reside directly on the substrate, including those described as demersal, benthopelagic, or bathydemersal (FB, SLB). The benthic category includes species that inhabit or are closely associated with the seabed, such as those described as benthic or reef-associated (FB, SLB).

- *Distance to coast* defines the geographical position of the habitat in the spatial dimension and the affinity of species to coastal and/or open ocean systems. The descriptor was computed using cleaned occurrence data (section Development of range section). Based on the Benthic Provinces of the World [25] (BPOW) depth zones, we computed the percentage of clean occurrences located in the coastal domain. If 90% of the records were in the coastal area, the species is described as neritic only, and if 90% of the data is located outside the coastal area the species is described as oceanic only. In any other case, the species is considered generalist.

- *Distance to surface* defines the geographical position of the preferred habitat in the vertical dimension that ranges from the surface to the deep ocean environment. Classification of this descriptor differs from the Distance to bottom descriptor. If the species is described as pelagic, we use the overlap of the depth ranges (buffered by 30 m) gathered from FB, SLB or literature and vertical division defined environmentally by Longhurst [26] and ecologically following the FB database. Three non-exclusive categories are defined here: *epipelagic* (0–200 m), *mesopelagic* (200–1,000 m), and/or *bathypelagic* (>1,000 m). If the species is defined as demersal and/or benthic, we relied on the vertical division proposed by Watling et al. [27] and described and adapted in BPOW as depth zones: *coastal & upper bathyal* (0–800 m), *bathyal* (800–3500 m), *abyssal* (3500–6500 m), or *hadal* (>6500 m). The species is then assigned to one or more vertical categories based on its depth range if available or, if unavailable, on the percentage of occurrence points (>10% of the records) in each BPOW depth range categories.

                                                                                      

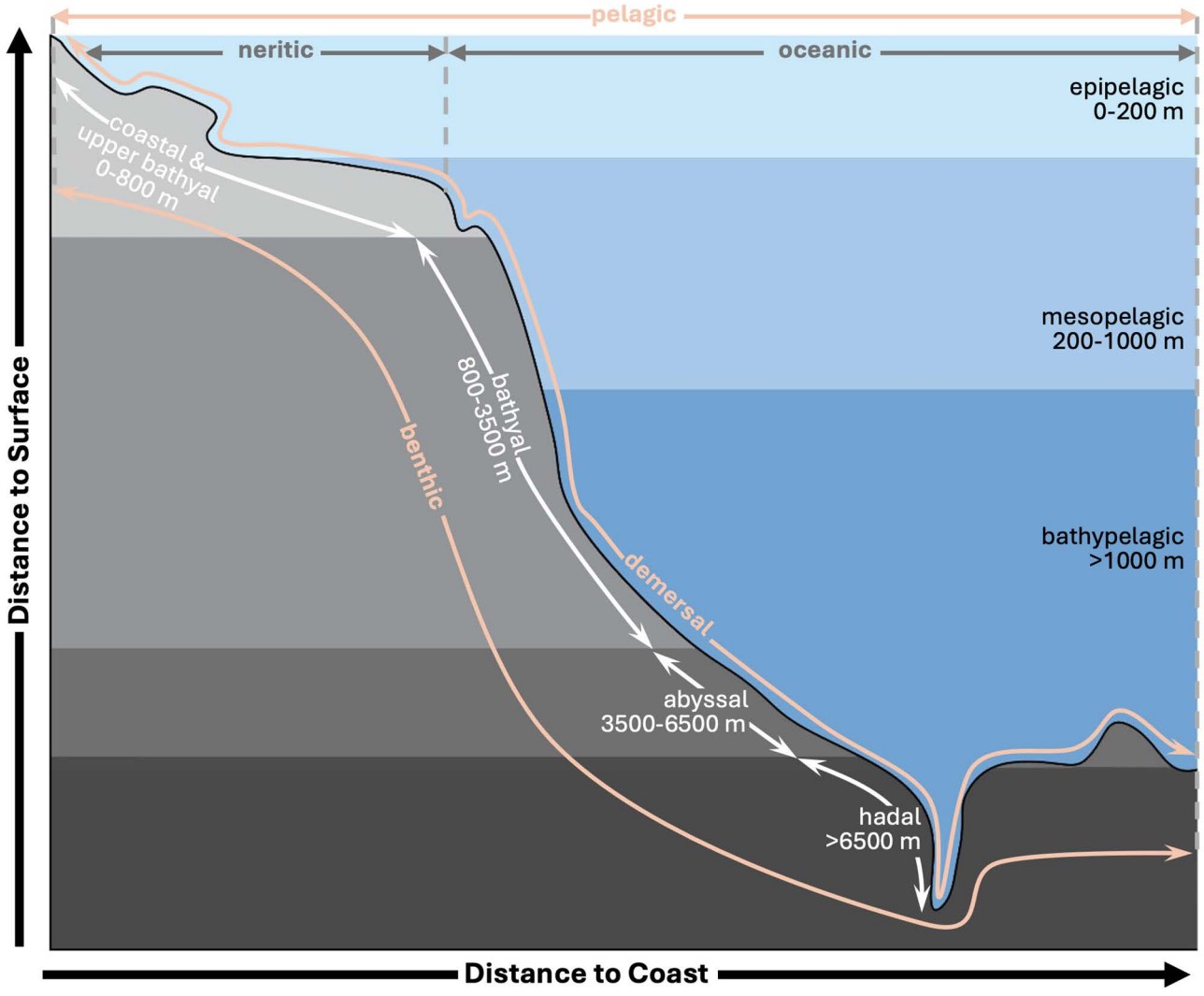

**Fig 1. Schematic representation of AquaX Distance-3 Ocean System (D3OS) classification.**

## Development of range extent

Range maps are alternatives to species distribution models that are typically created to represent both the current and potential future geographical boundaries of species distributions. These maps can be informed either by expert knowledge, such as those provided by the IUCN [24], or empirically by geographical delineation based on habitat preferences, species traits, taxonomy, and existing occurrence records (e.g., by drawing bounding boxes). In practice, range maps play a crucial role in mitigating common biases associated with species distribution models [28], such as the hemisphere or basin mirror effects, which can lead to projections into environmentally suitable but ecologically implausible habitats. We have integrated range maps into our workflow to help constrain AquaX projections and improve ecological realism:

- *Biogeographical Range* (BR): The area where a species is currently described either by an expert or with biogeographical confirmed evidence in an ecological division such as an ecoregion.

- *Potential Range* (PR): The area where a species could potentially occur in the future, based on the BR.

Most marine species lack an Expert Range Map (ERM), so we developed two complementary approaches to define species ranges. In the sections that follow, we first describe the method used when an ERM is available and then outline our approach for species without an ERM.

## Implementation of range maps using expert range maps

IUCN Expert Range Maps (ERMs) provided the basis for defining species range maps to help constrain SDM projections (for flowchart see S3 Fig). All available ERMs were downloaded from the IUCN database (downloaded September 2024), and species names were standardized using the AphiaID from WoRMS. ERM maps without an accepted scientific name in WoRMS were discarded. In cases where multiple ERM maps corresponded to a single valid WoRMS scientific name, maps were merged into a single representation.

Each ERM map was used as a BR with a buffer of 1° of longitude and latitude (to account for potential spatiotemporal uncertainties). They were subsequently used as a framework to generate PR by applying polygon buffering of 10° of longitude and latitude. The buffer was chosen to capture the mean poleward shift of marine species of $5.9 \pm 0.9$ km yr$^{-1}$ [29]. Since buffering can extend ranges into unrealistic locations, we applied an ocean mask to retain only areas within the same ocean basin as the original ERM. Additionally, to prevent the buffered range from overlapping land areas, we cropped out any landmass from both the PR and BR. We updated the existing flags in the occurrence dataset to mark points outside the BR as erroneous, i.e., points that were in global databases but outside of expert-derived range maps. Since PR and BR were cropped to exclude land, all occurrence records found on land were automatically flagged as erroneous and excluded.

## Implementation of range maps for species with no ERM using ecological traits and occurrence records

In the absence of an ERM, we defined a species' range map based on the best available information- namely occurrence records, habitat preferences, and bioregional classifications (for flowchart see S4 Fig). Having previously processed the occurrence data by applying initial quality flags, we further refined the dataset to identify and minimize undetected errors that may still have been present among unflagged records. To further improve data reliability, we first removed all erroneous occurrences (flagged as 0). We then applied AquaMaps bounding boxes, originally developed by the AquaMaps team based on species-specific environmental tolerances, habitat preferences, expert knowledge, and biogeographic data from FB and SLB. For species with multiple entries under the same accepted WoRMS scientific name, we merged their bounding boxes into a single representative range by selecting the most extreme latitudinal and longitudinal coordinates. Occurrence records falling outside the merged bounding box were re-flagged as 0. For species lacking an AquaMaps bounding box, we removed unreliable records (flagged as 0 or located on land) and generated a new square bounding box using the outermost occurrence coordinates. These bounding boxes were then used to spatially crop datasets in downstream analyses, reducing computational demands by limiting processing to each species' biogeographical extent.

Before generating a range map, we minimized the potential spatial bias by applying spatial thinning (for species with >1,000 occurrences) to reduce the impact of heavily sampled areas and ensuring a more even spatial distribution. Spatial thinning was performed using *GeoThinneR* package [30], which simplifies data by rounding spatial coordinates and applying a hashing method to retain representative points. The thinning distance was set to 30 km for datasets with >5,000 occurrences and 20 km for all other cases. The thinning distances were chosen to balance the reduction of spatial sampling bias with the retention of sufficient ecological signal, as supported by empirical testing and best-practice recommendations in spatial modeling tools [31,32]. In addition, A 30 km thinning justified by alignment with oceanographic gradients and typical daily displacements of marine organisms [33].

We then used this information to assign each species to ecosystem or biogeographical divisions where it had a confirmed presence or demonstrated ecological affinity, providing a basis for delineating species-specific spatial extents. Here, we assumed that the confirmed presence of a species within a biogeographical unit indicates its potential

occurrence throughout the entire unit, while acknowledging the non-homogeneous environmental gradients that typically characterize these regions. These biogeographical frameworks allowed us to delineate species distributions more accurately by aligning them with known ecological boundaries and habitat characteristics, reducing the likelihood of projecting species into unsuitable environments using ecological criteria. We used two main biogeographical partitions, the BPOW for benthic and demersal species and the Pelagic Provinces of the World (PPOW; Spalding et al. 2012) for pelagic species. The Arctic Marine Biodiversity Monitoring Plan (CBMP) Arctic Marine Areas [34] was used to subdivide the Arctic province of the PPOW and BPOW. This spatial resolution offers a compromise between macro-scale processes (realms/biomes) and finer ecological features (habitats/ecoregions). We have adapted this approach to 3 types of species: pelagic, benthic/demersal, and present in the Arctic.

For species classified as pelagic (according to D3OS), we first used bounding boxes (as described above) to define a broad spatial extent and then refined these preliminary maps using the PPOW at the province level. For pelagic-neritic species, we further restricted the range by cropping PPOW provinces using the BPOW to retain only coastal areas (0–800 m depth). To refine the resulting range maps, we applied a filtering approach based on occurrence frequency within PPOW provinces (after bounding box and depth filtering). For each species, we counted the number of occurrences per province, sorted them in ascending order, and calculated the cumulative occurrence percentage. Provinces were retained if they accounted for more than a predefined threshold (3.3%, approximately 1/30th of total occurrences). This step helped exclude provinces with very low occurrence density, which may reflect isolated or erroneous records rather than consistent species presence.

For species classified as benthic or demersal (according to D3OS), we applied the same initial bounding box approach to define the broad range and subsequently refined the distribution using BPOW. For reef-associated species, we further limited their extent to coastal BPOW areas to reflect habitat preferences. We then calculated the number of occurrences within each BPOW depth layer (coastal, bathyal, abyssal, hadal), sorted them by cumulative percentage, and retained only those layers contributing more than 3.3% of the total records. A similar filtering approach was applied to BPOW provinces, retaining only those accounting for more than 3.3% of occurrences. The final range map was defined as the intersection of the selected depth layers and provinces.

For species present in the Arctic, we applied an additional refinement step within the Arctic province. Since BPOW and PPOW each contain only a single Arctic province, they were insufficient for accurately delineating species distributions. To improve resolution, we overlaid the CBMP Arctic Marine Areas and assessed species occurrences within these units. We then calculated the percentage of total occurrences in each CBMP area and applied a 0.5% threshold (arbitrarily set) to exclude areas with minimal representation. The Arctic range was then redefined by retaining only CBMP units that exceeded this threshold, resulting in a more spatially detailed and ecologically plausible distribution within the Arctic region.

Once cleaned, each range map was used to compute BR and PR ranges by applying polygon buffering as described earlier. Since buffering can extend ranges into unrealistic locations, we applied an ocean mask to retain only areas within the same ocean as the original ERM and cropped out any landmass from both BR and PR. Then, we updated the existing flags in the occurrence dataset to classify records as valid if they were within the BR.

## Species modelling framework

**Environmental data.** Environmental data for modeling were obtained from Bio-ORACLE v3.0 [35] at a 0.05° resolution (~5 km at the equator) for present (2000–2010) and two future periods (2050–2060 and 2090–2100) under three SSPs: (1) SSP1–2.6, a low-emissions scenario targeting minimal greenhouse gas output, with a projected global mean sea surface temperature (SST) increase of 0.86°C (range: 0.43–1.47°C) from 1995–2014–2081–2100; (2) SSP2–4.5, a moderate-emissions scenario with a projected SST rise of 1.51°C (1.02–2.19°C); and (3) SSP5–8.5, a high-emissions scenario with a projected increase of 2.89°C (2.01–4.07°C) over the same period [36]. Predictor

variables were selected based on their biological relevance to the studied groups. Depending on species habitat, different depth layers were used. For pelagic species, surface data layers were used for modelling, while for benthic and demersal species, benthic data layers reflecting conditions along the sea bottom were used (Table 1). The same set of environmental variables used to estimate the current distribution of the examined species was also used to project their future distributions.

To determine the appropriate spatial resolution for analysis, we first assessed the proportion of modelled area in relation to the global ocean. Specifically, we calculated the number of cells in a given modelled region and compared it to the number of cells for the global ocean environmental dataset. Based on this ratio, we established a resolution adjustment criterion: if the proportion was less than or equal to one-third of the total oceanic cells, we selected a finer resolution of 0.05 degrees (~ 5 km at the equator). If the proportion was between one-third and two-thirds, we applied a medium resolution of 0.1 degrees (~ 10 km at the equator). For proportions exceeding two-thirds, we used a coarser resolution of 0.2 degrees (~ 20 km at the equator). This factor was used to refine or aggregate spatial data accordingly, ensuring an optimal balance between computational efficiency and spatial detail during the modelling process. To restore the final projection to its original scale (0.05 degrees), we applied a disaggregation process using bilinear interpolation, adjusting the resolution by the computed scaling factor.

**Preparation of the training set.** For each species, we first loaded verified species occurrence records from previous steps (i.e., using only records within BR flagged as valid) that were then spatially matched with respective environmental data based on their habitat.

Where an ERM was available, but the species had small occurrence number (<1k), we supplemented the data by generating pseudo-occurrences. The generation of pseudo-occurrences aimed to sample the environmental range of the species within its ERM that was not captured by observed occurrences, reducing niche truncation, and improving model projection. Pseudo-occurrence generation was further constrained to areas where bathymetry matched the species' known depth range; when no depth information was available, the entire ERM extent was used. This was done by rasterizing the ERM at 0.05° resolution and randomly sampling additional points equal to the minimum of either 1000 or 40% of the total number of raster cells within the ERM extent. To preserve information on which occurrences were auto-generated when saving data, we added a basis of record as "ERM".

SDMs can be affected by spatial biases in occurrence data, particularly when records are clustered due to variations in sampling efforts. To mitigate this issue, thinning was performed in environmental space rather than geographical space, as ENMs rely on ecological gradients rather than purely spatial distributions [7,8]. Geographic thinning may remove ecologically unique occurrences simply due to proximity, whereas environmental thinning ensures that retained records capture the full range of conditions the species occupies.

**Table 1. List of the Bio-Oracle (v3.0) environmental variables used for modelling fish species.**

| Habitat | Pelagic, Pelagic-oceanic, Bathypelagic, Pelagic-neritic | Reef-associated, Benthic, Demersal, Benthopelagic |
|---|---|---|
| Layer | Surface | Benthic |
| Dissolved oxygen concentration (mmol·m⁻³) | x | x |
| Ocean temperature (℃) | x | x |
| pH | x | |
| Primary productivity (mmol·m⁻³) | x | x |
| Salinity (psu) | x | x |
| Sea-water velocity (m·s⁻¹) | x | x |
| Average depth (m) | | x |

First, we determined a number of 'unique environments' for the global ocean based on binning of environmental variables used (Table 2), ensuring that values were rounded to meaningful increments. Once the raster layers were rounded, unique environmental conditions were identified based on distinct environmental combinations to represent globally unique environmental conditions. Each species' occurrence record was then matched to a 'unique environment' (i.e., multi-dimensional bin) based on its associated environmental conditions. Then, only one occurrence record was randomly selected per unique environmental location, ensuring a balanced representation of environmental conditions across the dataset.

**Pseudo absence generation.** Due to the need for information on both presences and absences for various SDM algorithms used in AquaX, pseudo-absence points were generated to complement the occurrence dataset, as is common in SDM approaches. The selection method of pseudo-absences (PAs) was adapted to the species habitat characteristics:

- For pelagic species, PAs were selected using the Surface Range Envelope (SRE) method (from 2.5 to 97.5 percentiles) [7,37]. Pseudo-absences were randomly selected from areas where all environmental variables fell outside the SRE within the PR, thus ensuring selection of locations within the PR with higher probability of absence.

- For benthic/demersal species, which are constrained by substrate type and depth, PAs were generated using a habitat-informed disk sampling strategy [38]. Pseudo-absences were generated within a distance range of 80 km to 1,000 km from known occurrences. Pseudo-absences were only generated within the PR of the species.

The pseudo-absence selection was restricted to the PR area, which serves as a biologically meaningful region for modeling. The PR area includes information both on areas where the species is currently present and areas where it is absent due to currently unsuitable environmental conditions. This approach allows (i) avoiding overfitting the models by picking absences only in regions that are located in the near or at the edge of the environmental niche of the species so as to not inflate the true negative in the validation process, (ii) by selecting sites where the probability of true negative is optimized by lowering the false positive. Consequently, these restrictions ensure that PAs are selected within a relevant ecological and geographic context, avoiding unrealistic placements in regions.

**Model algorithms and single model training.** ENMs or SDMs were implemented with the R *biomod2* package version 4.2–4 [37,39]. More specifically we used ten of the statistical algorithms available (i.e., providing the name of the method Aqua X): generalized linear models [40] (GLM), generalized additive models [41] (GAM), random forests [42] (RF), artificial neural networks [43] (ANN), flexible discriminant analysis [44] (FDA), classification tree analysis [45] (CTA), Generalized Boosting Model, or usually called Boosted Regression Trees [8](GBM), Maximum Entropy [46](MAXNET), Extreme Gradient Boosting [47](XGBoost), Multiple Adaptive Regression Splines [48](MARS).

Using the filtered and simplified training dataset, each model was individually tuned using the biomod-tuning command to optimize hyperparameters for each species. Following the tuning of the ten algorithms, when the number of unique environmental conditions were >1000, a five-fold environmental cross-validation strategy was employed to evaluate model

**Table 2. Binning of the global set of environmental variables.**

| Environmental Layer | Bin size |
|---|---|
| Dissolved oxygen concentration (mmol·m$^{-3}$) | 10 |
| Ocean temperature (℃) | 0.05 |
| pH | 0.1 |
| Primary productivity (mmol·m$^{-3}$) | 0.5 |
| Salinity (psu) | 0.1 |
| Sea-water velocity (m·s$^{-1}$) | 0.02 |
| Average depth (m) | 5 |

performance. Unlike traditional random cross-validation, this approach partitions occurrence data based on environmental space rather than geographic space, ensuring each fold represents distinct environmental conditions (CV.strategy = 'env' in *biomod2* package). The cross-validation process involved splitting the dataset into five environmental groups, training models on four folds, and evaluating them on the remaining fold. This procedure was repeated once for each environmental variable, resulting in 30 runs per species. Presence and pseudo-absence data were balanced within each fold, ensuring equal weighting of occurrence information. The prevalence of presence data was set at 0.7, thereby prioritizing presences over PAs during model training. Consequently, this resulted in 30 modelling runs per algorithm, amounting to a total of 300 individual model runs per species. For species with low numbers of unique conditions we did repeated 5-fold cross validation (3 repeats) as some validation repetitions do not have both presences and absences.

**Evaluation metrics.** A combination of threshold-dependent and independent metrics was used to evaluate the models. Models were calibrated using the area under the curve (AUC) of the receiver operating characteristic, true skill statistic (TSS) and critical score index (CSI). TSS and AUC were used to evaluate model performance using the *pROC* package [49]. TSS is defined as the sum of sensitivity (true positive rate) and specificity (true negative rate) minus one [50]. The AUC is a threshold-independent measure representing the relationship between sensitivity and the corresponding proportion of false positives (1 – specificity). AUC ranges between 0 and 1, with values above 0.9 indicating excellent prediction and values below 0.5 indicating a projection no better than random AUC [51]. CSI measures the proportion of correctly predicted presences relative to the total number of observed or predicted presences, accounting for both false positives and false negatives. CSI is particularly useful when working with presence-pseudo-absence data, as it provides a robust evaluation of model accuracy in imbalanced datasets and does not rely on true negatives, making it more suitable for cases where absence data are uncertain or artificially generated.

**Ensemble modelling.** Following the evaluation of single models, we only kept models with TSS values exceeding 0.6 for the ensemble approach. These models were then combined using the *biomod2* package to create the final ensemble model [52]. To generate ensemble predictions, we applied three integration methods: mean ensemble, committee averaging, and the coefficient of variation. The mean ensemble model computes the average probability of occurrence across all individual models, providing a continuous suitability estimate. Committee averaging applies a majority rule approach, where a species is predicted as present if more than half of the models agree, resulting in a binary output. Finally, the coefficient of variation quantifies uncertainty in model predictions by measuring the relative dispersion of probability estimates, highlighting areas where models exhibit the highest disagreement.

Upon completing the ensemble modelling, we used the generated model to map the potential habitat of species under present and future scenarios. Future climate scenarios were characterized by three SSPs (SSP1–2.6, SSP2–4.5, and SSP5–8.5) and were considered for two-time frames: the 2050s and the 2090s. For each time period, the model was executed individually, utilizing future bioclimatic variable data.

## Results and discussion

### On the evolution of AquaMaps methodology

The original idea for AquaMaps to approximate the global distribution of marine mammals using a set of observations and environmental data was developed in 2006 [13]. Based on the evolution of a Surface Range Envelope that simplified the ecological niche *sensu* Hutchinson [12], Kaschner *et al.* developed a trapezoidal model (named Relative Environmental Suitability) that approximated the relationship between a marine mammal species and the environment. Furthermore, it also projected the spatial realization of the simplified niche in current conditions based on expert knowledge on species-specific habitat preferences for three environmental parameters (depth, sea-surface temperature, and ice edge association) available at the time (at 0.5° of spatial resolution). The model was subsequently refined by Ready *et al.* [53], who automated the approach for a continuously expanding taxonomic array of marine species by incorporating occurrence records harvested from increasingly more available online data portals supplemented by additional information about

habitat usage from species databases such as FB and SLB. Here, we propose a transparent, fully replicable framework based on all the most recent methods and database available to improve AquaMaps through the new AquaX framework, introducing four major new updates:

1. Full replicability through reliance on the freely available software R,

2. the use of higher-resolution environmental data based on Bio-Oracle v3.0,

3. a novel methodology designed around ecological, biogeographical, and taxonomic traits of marine species,

4. an ensemble modeling approach comprising ten independent models to provide long term annual average distributions

The first update of the AquaMaps methodology was the development of a workflow fully replicable in R and based on open-source databases such as WoRMS, OBIS, GBIF, FB, SLB and IUCN. The use of the AphiaID for each species is a step toward a normalization of descriptors such as taxonomy, occurrence or ecological/biological traits across platforms to simplify the search for users worldwide. All meta information used by the AquaX methodology can be retrieved using R and the AphiaID (except for GBIF and IUCN).

The second update involves the use of Bio-Oracle v3.0 [35] as a standardized environmental database for model training and projection. The selection of this database aligns with the previously described rationale, emphasizing replicability and supporting international collaboration. Bio-Oracle v3.0 offers environmental data at an improved spatial resolution of 0.05° latitude and longitude, enabling more precise characterization of local and regional environmental features. In the spirit of building up from the previous work of AquaMaps, AquaX offers an updated species distribution at 0.05° grid (10 times finer than AquaMaps). This higher resolution is especially noteworthy as it more accurately represents the biogeochemical and physical attributes of coastal ecosystems where most marine habitats and biodiversity occur [54,55]. Furthermore, such enhanced resolution is crucial for modeling smaller-scale features such as smaller marine protected areas and coastal zones, where spatial features often fall below the detection threshold of traditional coarse-resolution models (typically 1° or 0.5°), making it especially relevant for current and future conservation planning [56].

The third major update proposed by AquaX involves the implementation of a novel methodology designed around the ecological, biogeographical, and taxonomic traits of marine species, aimed at defining or refining realistic species habitat suitability maps. This approach utilizes established biogeographical divisions informed by ecological characteristics and/or expert delineations to establish geographical boundaries as foundational inputs for the subsequent ensemble modeling. While this methodology inherently incorporates assumptions, it explicitly addresses common biases such as uneven sampling effort and species misidentification [53]. However, as all outcomes involving such assumptions should be subject to scrutiny, the framework provided by AquaX encourages and calls for active collaboration. Specifically, it facilitates global expert engagement by offering numerical frames designed to update existing range maps and to define new distribution ranges, thereby addressing critical gaps frequently encountered in the assessment and conservation planning of marine biodiversity [15].

The final major update of AquaX employs an ensemble modeling approach comprising ten independent models, inspired by methodologies utilized in the FISH-MIP and CMIP projects [57,58]. Each model was specifically selected to encompass various types of ENM and species distribution modeling (SDM) techniques, including linear models (GLM, GAM, MARS), background or entropy-based models (MAXNET), explorative statistical approaches (CTA, RF, FDA), and machine learning models featuring pruning and boosting techniques (ANN, GBM, XGBOOST). This combination of diverse modeling approaches aligns with best practices recommended in terrestrial ecology [39,59,60] to effectively capture the multidimensional complexity inherent in ecological niches. The ensemble output, despite potentially overestimating niche boundaries and underestimating central niche probabilities due to averaging effects [61], provides an optimal compromise, especially when projecting under future environmental scenarios.

To illustrate the methodological advancements and projections of species' habitat suitability, we present a comparative analysis of bigeye tuna (*Thunnus obesus;* Lowe, 1839) habitat suitability, contrasting current suitability outputs from AquaMaps (Fig 2A) at a resolution of 0.5° against the updated AquaX ensemble model at an enhanced resolution of 0.05° (Fig 2B, C). Individual outputs from each contributing model are provided in supplementary materials (S5 Fig). AquaX includes spatial and environmental uncertainty estimates to evaluate the consistency and predictive accuracy of the ensemble model outputs (Fig 2D, E). Spatial variability among models is assessed through the coefficient of variation across ensemble projections, revealing areas of disagreement and lower confidence in predicted suitability (Fig 2D). Environmental extrapolation is identified through maps showing where environmental conditions fall outside the range used for model calibration, indicating potential limitations in predictive reliability (Fig 2E). In parallel, spatial variability among models is assessed through the coefficient of variation across ensemble projections, revealing areas of disagreement and lower confidence in predicted suitability. Together, these outputs provide a spatially explicit representation of uncertainty within our modeling framework. Such estimates can guide further validation efforts and provide a framework for cautionary interpretation of projections. Additionally, each individual model run, along with the overall ensemble model, underwent rigorous validation using three independent performance metrics, ensuring transparency and enabling users to assess the reliability of the models before application (S6 Fig).

## Applications of AquaX

**A tool for expert-based spatial delineation of species ranges.**  AquaX has been developed to support ecologists, conservation planners, and biodiversity researchers aiming to establish or refine species ranges and study biodiversity distributions. AquaMaps, the basis of AquaX, has long provided a foundational resource for marine conservation planning, supporting efforts such as the identification of priority areas for protection, Marine Protected Area (MPA) siting, and climate risk evaluation. Numerous studies have leveraged AquaMaps outputs to inform the design and adaptive management of MPAs, integrating species distribution models into conservation strategies to anticipate the impacts of climate change and biodiversity loss. The refined AquaX framework, with its higher spatial resolution, ensemble modeling, and robust validation metrics, offers even greater utility for conservation and management applications. By enabling more accurate projections of species' habitat suitability under multiple climate scenarios, AquaX can help managers and policymakers identify climate-resilient sites, optimize MPA networks, and evaluate ecological risks, while acknowledging the inherent uncertainties and limitations of modeling approaches. As climate adaptation becomes increasingly central to marine spatial planning [62,63], the advances in AquaX position it as a powerful tool for supporting evidence-based decision-making in a rapidly changing ocean.

AquaX can also serve as a foundation for refining modeling strategies, allowing users to integrate additional environmental or biological parameters to enhance predictive accuracy (code and data will be public). Furthermore, this approach can aid taxonomic and biogeographic experts by streamlining the process of creating species range maps. By generating an initial species distribution that aligns with environmental constraints, experts can focus on refining and validating these predictions rather than manually delineating ranges from scratch. This significantly reduces the time and effort required to assess distributions for unassessed species, making the approach an efficient tool for initial decision-making in biodiversity conservation and resource management. Furthermore, the iterative improvement process described above may also help to further refine and integrate both range maps and SDM projections.

**An update of AquaMaps single species current and future distribution at high resolution.**  The new AquaX framework was run for all species with sufficient occurrence data, which presented with different ecology (i.e., benthic vs pelagic vs pelagic-neritic), allowing to unveil the advantage of the methodology especially for coastal species. We combined all outputs of the methodology to present an example of the product that will be available to users. Fig 3 illustrates the output for the sand seatrout, *Cynoscion arenarius* Ginsburg, 1930. As the record of the sand seatrout occurs not only within the ERM but also extends northward along the coast of Georgia, North and South Carolina (Fig 3A),

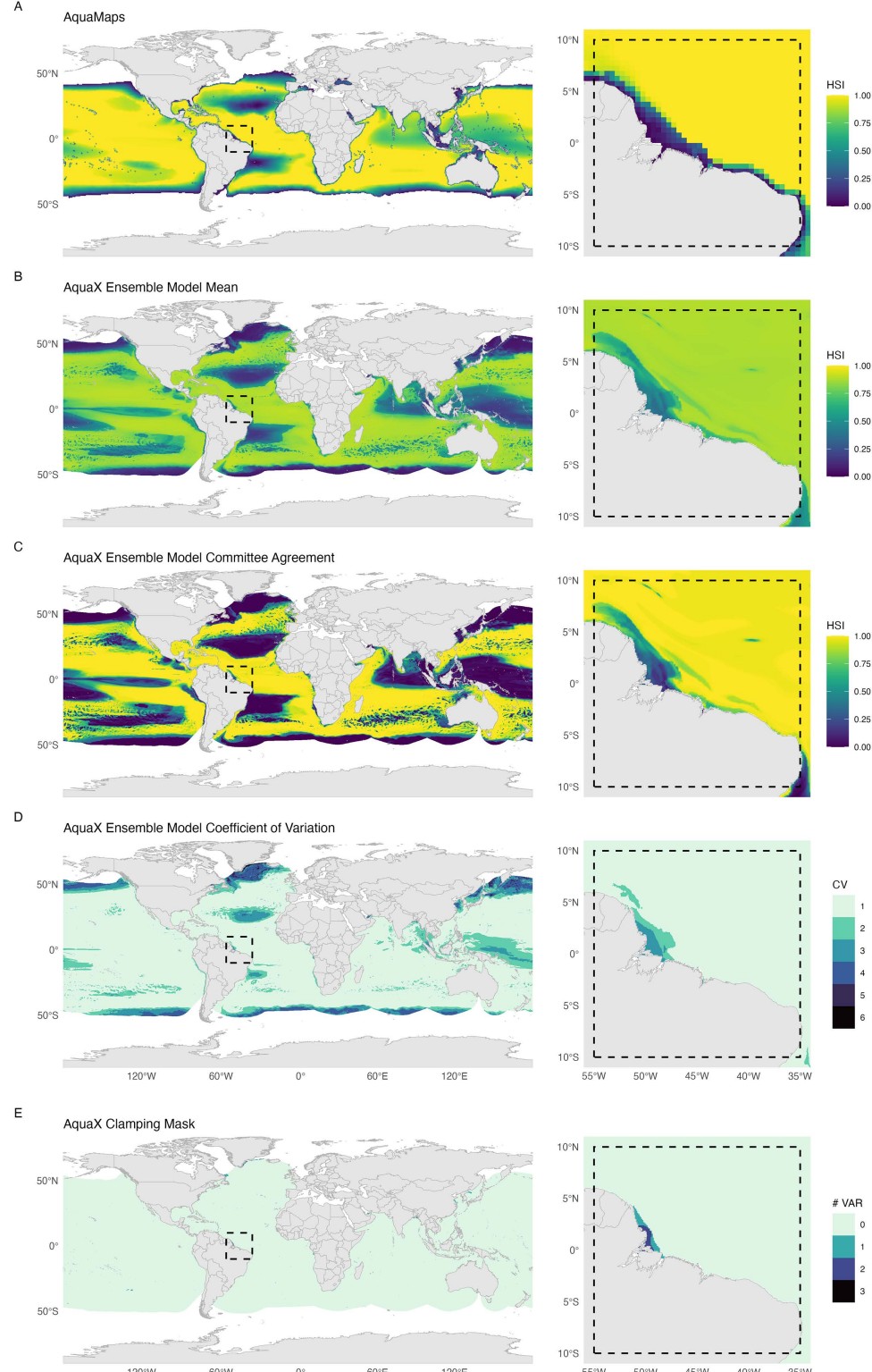

**Fig 2. Maps of the big eye tuna, *Thunnus obesus* (Lowe, 1839), following the AquaMaps and AquaX methodology.** A) Habitat Suitability Index (HSI) for the most recent version of AquaMaps. B) HSI for the AquaX ensemble mean. C) Committee Average (CA) for the AquaX Ensemble model. A value of 1 indicates that all the runs agree on the presence (based on binned HSI) of the species while 0 all runs agree that the species is absent. D)

AquaX Ensemble Model Coefficient of Variation (EMcv). A higher EMcv value indicates greater disagreement among models, while a lower value means more consistency. E) Clamping mask showing the number of environmental variables whose value is outside the range of values used to calibrate the models. Maps in the second column show a zoomed-in section of the corresponding maps in the first column (dashed rectangle). Made with Natural Earth. Free vector and raster map data @ naturalearthdata.com.

we used the AquaX methodology within the PR (Fig 3B) to re-evaluate the range of the species using the BR and PR approach (Fig 3B). We subsequently ran the AquaX ENM to quantify within the PR the current suitable habitats of the species at 5 km, showing that records outside of the ERM were located in highly suitable areas for the species (Fig 3C and D). This finding suggests a potential poleward shift in the distribution of *C. arenarius*, possibly driven by environmental changes such as ocean warming or other environmental variables. Alternatively, these records may represent previously undocumented occurrences that were not captured during the initial creation of the ERM, highlighting the importance of dynamic range reassessments integrating methodologies like AquaX.

Based on the validated ensemble model, we can subsequently explore the climate-driven change in the distribution of the species following various SSP–Representative Concentration Pathway (RCP) scenarios and time frames. Changes in the HSI between future projections (2050s and 2100s) and present conditions (2000s) at high resolution provide critical insights for potential management and conservation purposes. A decline in HSI over time may indicate habitat degradation and climate-driven range contractions, signaling areas where conservation actions may be needed. Conversely, an increase in HSI suggests habitat improvement potentially causing climate-driven range expansions or adaptive responses of species. By comparing future and present HSI (Fig 3E, F), researchers and policymakers can anticipate changes in species distributions, assess ecosystem resilience, and implement proactive conservation strategies. For example, in the case of sand trout, an increase in HSI is projected in its northward range by mid-century (Fig 3E, F), along with a potential shift to deeper waters (blue areas below a red strip in Fig 3F).

The potential areas in species invasions and extinctions can be summarized in a simple representation (Fig 4). Using HSI maps (Fig 3), we can convert continuous habitat suitability values into binary presence-absence maps by applying a threshold (provided by our methodology, *biomod2*) and compute climate driven indices (e.g., local extinctions, invasions). Local invasion refers to the expansion of a species beyond its historical range, driven by environmental changes. Local extinction, in this context, signifies the disappearance of a species from previously suitable habitats due to change in environmental conditions. By analyzing these binary maps over time (Fig 4), we can compute the spatial extent of habitat loss and gain, providing not only insights into habitat degradation or improvement but also into actual species distribution dynamics for policy makers and stakeholders.

High-resolution HSI mapping is especially essential for local-scale management of endangered populations as it provides more precise spatial information, allowing for targeted and effective conservation actions [64–66]. A finer resolution enables the identification of critical microhabitats, small-scale habitat fragmentation, and localized environmental changes that may be overlooked in coarse-resolution models [62,67]. This level of detail is particularly important for species with narrow ecological niches or those sensitive to habitat degradation, as it ensures that conservation strategies address specific threats at the appropriate scale [68,69]. Additionally, high-resolution data facilitate more accurate climate change impact assessments, habitat connectivity analyses, and area-use planning, enabling policymakers to make informed decisions for habitat restoration, protected area expansion, and species management. By capturing fine-scale variations, high-resolution HSI mapping supports adaptive conservation strategies that can be tailored to regional and local needs, ultimately improving the long-term survival of endangered species.

**A new tool to assess global biodiversity.** A significant advancement provided by AquaX lies in its applicability for macroecological studies. To illustrate the advantages of AquaX over previous methodologies, we conducted a comparative analysis of coral species richness along Australia's Great Barrier Reef (Fig 5A–C). The species richness estimated by AquaX was first compared with high-resolution information from the Coral Reef Atlas produced by the World Conservation

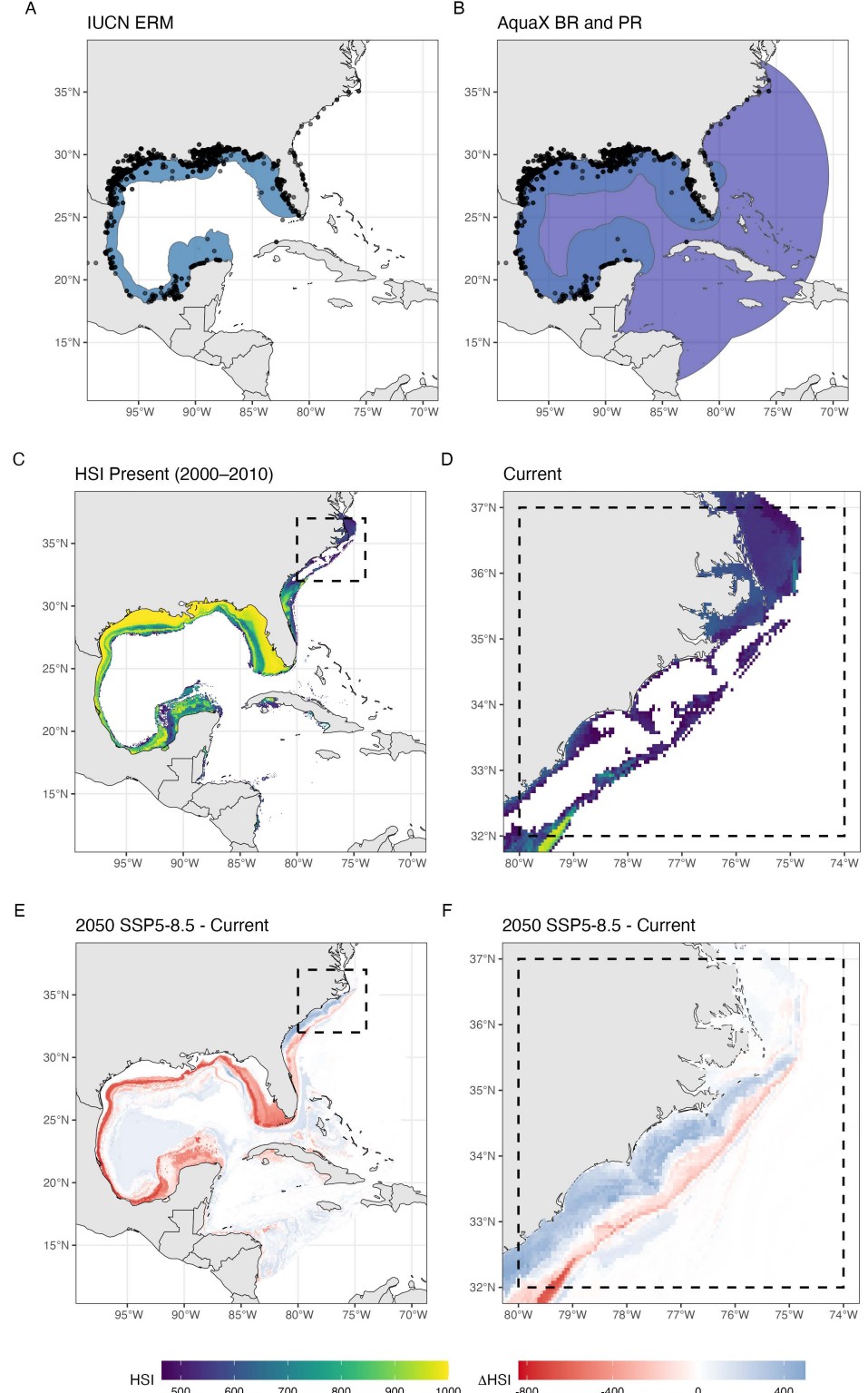

**Fig 3. Distribution and habitat suitability of sand seatrout (*Cynoscion arenarius*, Ginsburg 1930).** (A) IUCN expert-defined range map. (B) Bio-geographical (blue) and potential ranges (purple) from AquaX. (C) Habitat Suitability Index (HSI). (D) Zoomed-in portion of (C) highlighting the northern range of the species. (E) Change in HSI between future projections (SSP5–8.5, 2050–2060) and present conditions (2000–2010). (F) Zoomed-in portion

of (E) highlighting the northern range of the species. Black dots are occurrence points. Made with Natural Earth. Free vector and raster map data @ naturalearthdata.com.

Monitoring Centre [70] (WCMC). The comparison revealed that AquaX captured the high-resolution location data provided by the Atlas. Furthermore, AquaX-derived species richness value was compared with the existing similar database available at 1° resolution from Jenkins and Van Houtan (2016). We performed several geostatistical tests based on a resampling using maximum value of AquaX at the resolution of Jenkins & Van Houtan [71]with the *terra* package. First, we analyzed the spatial autocorrelation and distribution of variance between the two datasets and found a high similarity in the spatial gradient. Second, we conducted a linear correlation analysis and mapped residual errors across these datasets (Fig 5E and F). Two primary biases influenced the linear correlation between AquaX and the Jenkins & Van Houtan datasets: (i) a significant difference in spatial resolution (approximately a 20-fold difference), complicating direct comparisons despite a clear linear species gradient observed in the residual maps, and (ii) discrepancies in the species pools between AquaX and Jenkins & Van Houtan, leading to potential differences in spatial gradients of species richness [72]. Nonetheless, our analysis demonstrates that AquaX effectively replicates ($R^2 = 0.7$, p-value <0.05; linear model) broad-scale coral species richness patterns previously identified by Jenkins & Van Houtan [71]. Consequently, AquaX may substantially enhance species richness mapping, providing ecological detail comparable to fine-scale atlas datasets but for a broader range of species and offering a novel product to study the various components of biodiversity at high resolution.

## Usage & limitations

Our modeling approach provides a structured and adaptable framework for species habitat suitability modeling, offering a valuable starting point for researchers and practitioners seeking to refine their methodologies. By implementing a standardized workflow that incorporates environmental thinning, pseudo-absence selection, ensemble modelling and robust evaluation metrics, AquaX enables users to generate reliable species habitat suitability models while minimizing bias and computational inefficiencies. However, despite best efforts to minimize the impact of several biases, users need to be aware of some of the limitations of the AquaX framework as listed here in the framework sequence:

**Availability of ecological and traits information.** The challenge in collecting comprehensive ecological information for marine biodiversity varies significantly across different taxa. While emblematic and commercially exploited species, which often include vertebrates and prominent biogenic habitats, typically have well-documented habitat preferences and depth ranges, considerable gaps remain for many other taxa. Specifically, our database indicates that around 22% of fish species lack habitat and depth information (S7 Fig). Additionally, the majority of marine invertebrates, which constitute the bulk of marine species richness, suffer from a pronounced lack of detailed ecological metadata and approximately 85% of mollusks species in our dataset do not have associated habitat or depth range information (S8 Fig). Consequently, our team addressed existing knowledge gaps by assigning missing habitat information based on the most common habitat type observed within the genus or, if genus-level data were unavailable, within the broader taxonomic family. Additionally, we updated incomplete depth ranges by inferring depth information directly from occurrence data verified through our methodology. These occurrence-derived depth values provided precise depth ranges, ensuring accuracy and ecological relevance for each species. However, these depth ranges are data-driven and are currently being verified (species by species) by FB and SLB experts.

This disparity underscores a significant bias in biodiversity research and conservation efforts, akin to that in the terrestrial realm, highlighting the urgent need for targeted data collection to better represent invertebrate taxa (DigIn project in USA, Digitizing Marine Invertebrate Collections), particularly given their ecological importance and diversity [73,74].

**Limitations of available expert range maps.** In the AquaX framework we value the evaluation of experts and consequently assume that ERMs represent a reliable framework for our occurrence verification process. However,

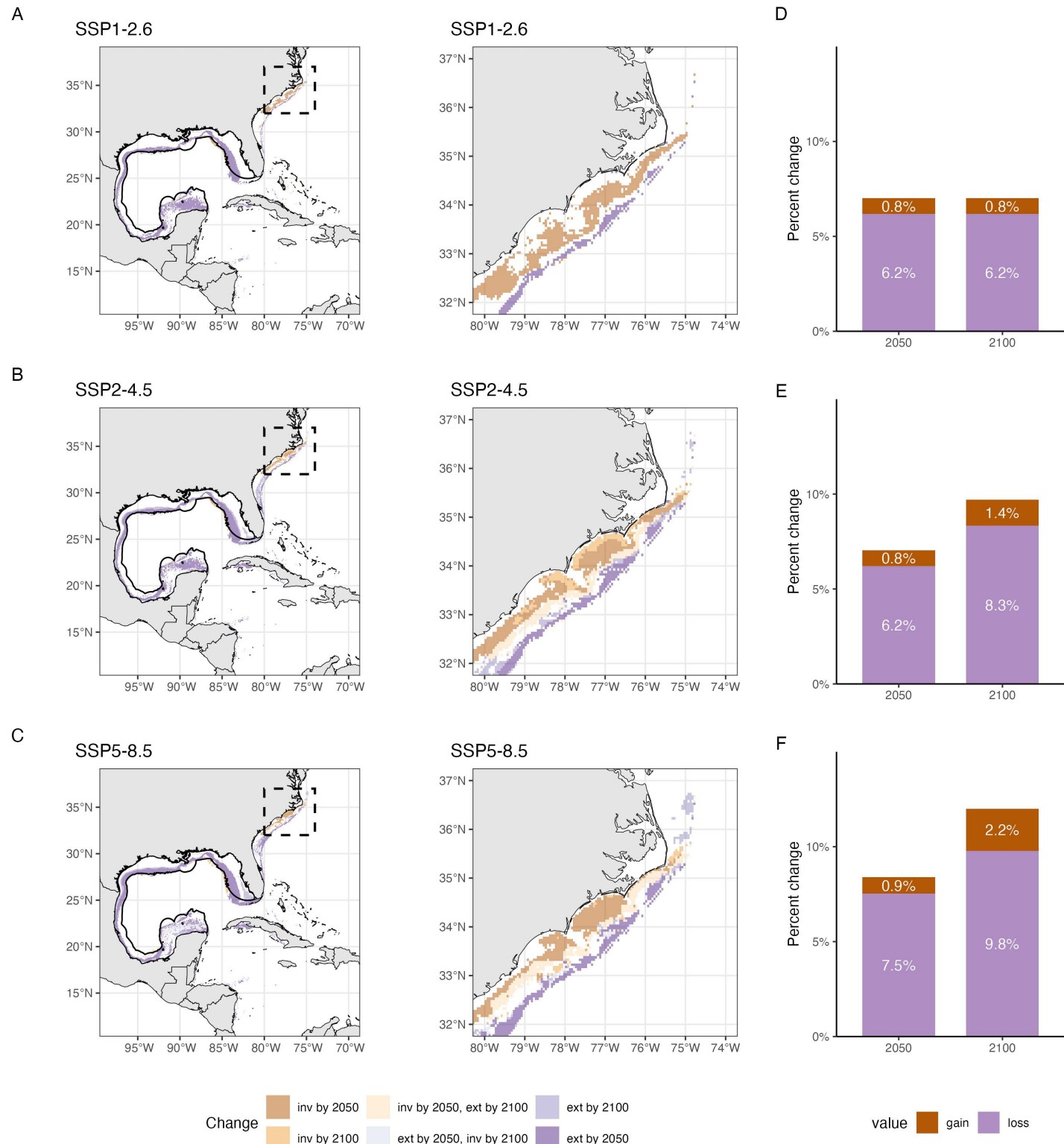

**Fig 4. Pattern of invasion (inv) and extinction (ext) of sand seatrout, *Cynoscion arenarius* (Ginsburg, 1930), for the middle (2050s) and end (2090s) of the century.** Map of invasion (species appears in areas where it was previously absent) and extinction (species disappears from the areas where it was previously present) of sand trout (and a zoomed in portion of the map shown in dashed line) under three climate change scenarios (A) SSP1–2.6 (B) SSP2–4.5 (C) SSP5–8.5. Solid black line delineates the boundary of the IUCN expert range map. Bar plots showing the percent lost or

gained of the modelled area (potential range from Fig 3B; total area of 3,988,793 km²) for the middle and end of the century under three climate change scenarios: (D) SSP1–2.6 (E) SSP2–4.5 (F) SSP5–8.5. Made with Natural Earth. Free vector and raster map data @ naturalearthdata.com.

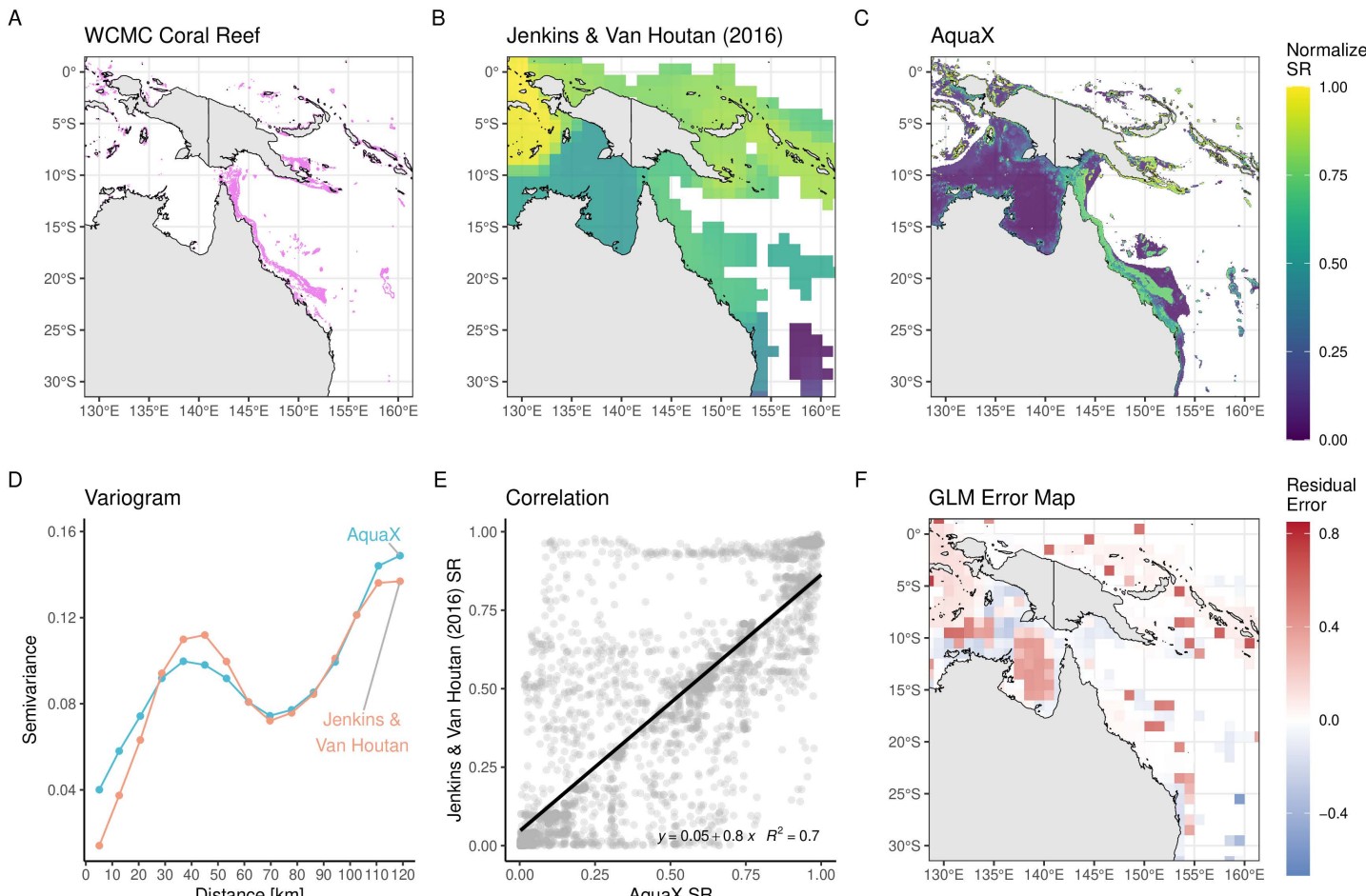

**Fig 5. Maps of coral diversity (normalized) around the Great Barrier Reef (GBR).** A) Distribution of warm-water coral reefs around the GBR region from UNEP-WCMC (1 km resolution). B) normalized coral species richness (SR) map from Jenkins & Van Houtan (2016, 100 km resolution). C) normalized SR map produced using AquaX (5 km resolution). D) Variogram for two coral datasets (global). AquaX dataset was resampled using maximum value to the same resolution of Jenkins & Van Houtan. E) simple linear regression between resampled coral SR (global) from AquaX and Jenkins & Van Houtan. Black line is linear regression. F) Map of residuals of linear regression. Made with Natural Earth. Free vector and raster map data @ naturalearthdata.com.

not every species or genus evaluated by IUCN has the same availability of information, which can result in an approximation of the range based on the knowledge of the expert in charge of the species. Unfortunately, this may result in overestimation or underestimation of the real ecological habitat of the species. For example, the expert range map of *Hippocampus barbouri* (Jordan & Richardson, 1908), a seahorse species with a depth range 0–12m (from FB), overlaps with the lower bathyal, abyssal, and hadal zones of the ocean (depth > 800m) (S9 Fig). Consequently, since the AquaX methodology relies on the ERM, erroneous occurrences might be kept (occurrences with a depth > 12m) and an overestimation of the distribution of the species in deeper waters will be made.

Furthermore, IUCN ERMs are updated at intermittent intervals, meaning that some species may have performed pole-ward migrations and hence the available range map may not accurately reflect the present-day distribution. For instance, numerous verified points of the snake pipefish (*Entelurus aequoreus,* Linnaeus, 1758) are situated beyond the species' known range map (S10A Fig highlighted in blue), with several of these points, marked in yellow, suggesting a potential northward range expansion. Despite using a buffered version of ERM to try to mitigate this issue, it fails to encompass all confirmed species locations. (S10B Fig).

Our framework attempts to balance these sometimes competing challenges and leverages expert knowledge and ensemble modeling outputs to refine existing current knowledge on range maps. This also works for species with limited occurrence data or when expert input is unavailable or sparse and can also be updated as and when such expert input or ERMs become available. We therefore advocate for an iterative approach wherein experts initially provide a preliminary distribution estimate, which is then tested and refined through our framework and subsequently reviewed and adjusted again by the same experts. This iterative process can significantly reduce errors and uncertainties inherent in both ERMs and SDM outputs. Previous research highlights that expert-derived range maps frequently contain inaccuracies due to subjective biases and incomplete ecological knowledge [53,75,76]. Thus, our proposed iterative framework represents an effective compromise, leveraging both quantitative modeling and expert judgment to generate reliable and scientifically robust range maps.

**Database occurrence biases.** Marine occurrence data from OBIS and GBIF are essential for the AquaX framework but suffer from significant issues, including duplication, inconsistent or outdated taxonomy, and spatial inaccuracies, despite considerable efforts to flag erroneous records, with most of the significant issues coming from the original database and provider. Cheung and Helfer [77] underline the persistence of erroneous records and stress improved validation and expert collaboration. Moudrý and Devillers [78] report that 35–55% of records have errors such as incorrect geographic coordinates or missing collection dates. Bosch et al. [79] further note the variability in OBIS data quality due to diverse data sources, emphasizing the need for thorough verification. Watts et al. [80] highlight the necessity of quality control to reduce duplication and enhance data publication standards, and Bonnet-Lebrun et al. [81] identify sampling biases limiting the use of marine occurrence data in ecosystem assessments. Avoiding incorrect occurrence records is essential for the AquaX framework as the species niche is built on the available occurrence data. Incorrect environment niches therefore can skew results of the modelling process and produce particularly problematic projections. An important aspect of our methodology deals with verifying occurrence data. This step is critical to ensure that only high-quality occurrence points are included in the model in order to more accurately specify the environmental niche of the species.

Verified occurrence data are also essential for constructing the BR and PR, which are used to constrain the ENM and future projections. For many taxa, ERMs are scarce, making it necessary to rely on available occurrence data and habitat information to infer potential species distributions. Having a well-verified set of occurrence points is crucial in such cases. For example, in fish–a well-studied group with 18,826 recognized species according to WoRMS–only 37% (6,923 species) have an associated IUCN Expert Range Map (ERM), while the remaining 63% lack expert-confirmed distribution data and verification of their occurrences. As additional species are reviewed and have ERMs generated by the IUCN, we can expect further constraining and improvement of the AquaX projections.

A significant portion of the AquaX methodology revolves around carefully selecting and verifying occurrence data. After downloading fish occurrence data and removing duplicates, on average, species contained 32.2% verified occurrences (from expert and FB/SLB verifications), and 5.4% erroneous occurrences and 62.4% of data having no flags. The AquaX data-cleaning approach further improved record verification. Species contained, on average, 58.2% verified occurrences and 24% more occurrences flagged as erroneous (and subsequently excluded), leaving only 17.8% unflagged.

To illustrate the data cleaning process, we present the Slinger seabream (*Chrysoblephus puniceus*; Gilchrist & Thompson, 1908) as an example (Fig 6). The raw occurrence data contained many unflagged records, including erroneous points located on land (Fig 6A). At the same time, some verified occurrences fell outside the ERM (Fig 6D), indicating

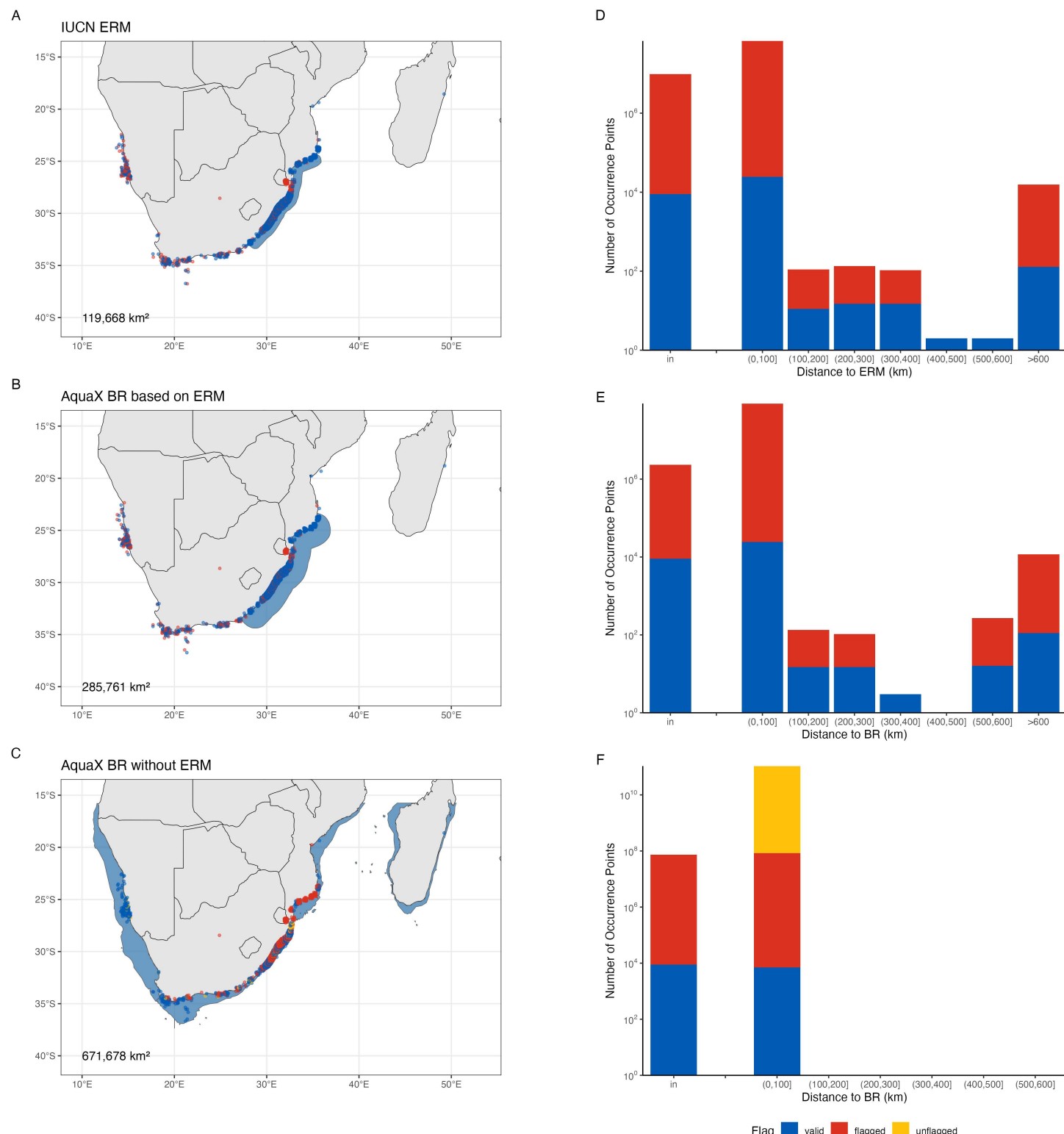

**Fig 6. Range maps and occurrences data of Slinger seabream, *Chrysoblephus puniceus* (Gilchrist & Thompson, 1908).** Erroneous (flagged) occurrences are marked by red, verified (valid) occurrence points are marked in blue, records with no flags (unflagged) are marked as yellow. A) raw occurrence data and IUCN ERM (shaded in blue shade). B) verified occurrence points and biogeographical range (BR) map (in blue shade) produced

from AquaX methodology. C) verified occurrence points and BR map (in blue shade) produced from AquaX methodology assuming no ERM available. Numbers in the bottom left corner of the maps are the total area of the range maps in square kilometers. Distribution of flags with the distance from D) ERM, E) BR built based on ERM, and F) BR built without ERM. Made with Natural Earth. Free vector and raster map data @ naturalearthdata.com.

potential discrepancies between the ERM and actual species distributions as a confirmed species occurrence was recorded near Madagascar or along the south coast and west coast of Africa. Unflagged occurrences were scattered throughout the species' range, highlighting the need for a systematic approach to verifying these data points. Using the AquaX procedure, we constructed a BR based on the ERM while systematically verifying unflagged occurrences. Assuming the ERM accurately reflected the species' true distribution, occurrences located within the ERM boundaries were marked as verified, whereas those outside these boundaries were flagged as erroneous (Fig 6B, E).

This method, however, has some limitations. Certain occurrence records outside the BR may still represent valid occurrences, as observed with the seabream species, where the ERM does not fully encompass the known native range of the species. Consequently, expert revision remains crucial to correctly verify and validate these occurrence points. We consequently inform IUCN and OBIS/GBIFF about these records.

Additionally, we observed an increase in occurrences flagged as erroneous close to the ERM and BR boundaries. This primarily results from insufficient spatial resolution of the coastline data used in the study, causing the BR to inadequately represent nearshore environments. Therefore, careful consideration of coastline resolution and incorporation of expert validation are essential to minimize these limitations. In cases where a species lacks an ERM, the AquaX methodology generates a BR using available occurrence data and habitat information (e.g., Fig 6C). Occurrences within this BR are marked as verified, while unflagged points located outside remain unverified (Fig 6F). This information is sent back to the original website (OBIS/ GBIF). This approach retains more verified data, thereby increasing the occurrences available for modeling. However, this can potentially introduce several sources of bias: The limitation of excluding the ERM is that it produces excessively broad predicted ranges, which, although capturing a higher number of occurrence points, may significantly overestimate the species' true ecological distribution, thereby reducing model precision and potentially affecting conservation and management effectiveness. On the other hand, some ERM can be considered too narrow and underestimate the "real" range of the species. In our framework, we chose to rely on expert opinion and use ERM when available; while we recognize that ERMs are not ideal, we plan to refine and improve them in the next iteration of the model. Furthermore, occurrences situated within approximately 100 meters of the BR boundary, especially in complex coastal regions, may remain unflagged due to the truncation of nearshore habitats. This is due to insufficient spatial resolution of coastlines, which cannot adequately capture narrow channels and complex coastal features, leaving certain nearshore occurrences unflagged.

**Limitations of available environmental layers.** A key limitation of our modelling approach stems from the structure of the Bio-ORACLE v3.0 dataset [35], which provides environmental data only for two vertical layers—surface and bottom. While this framework supports a broad classification of pelagic and benthic/demersal habitats, it does not capture the vertical heterogeneity of the water column, such as conditions in the mesopelagic or bathypelagic zones. This simplification may limit the accuracy of projections for depth-stratified species. However, this reflects a deliberate trade-off in Bio-ORACLE v3.0, which prioritizes high spatial resolution (0.05°) and global coverage over vertical complexity. Despite these constraints, Bio-ORACLE v3.0 remains one of the most widely used and comprehensive datasets for marine species distribution modelling, offering a rich set of biologically relevant variables and consistent future projections aligned with CMIP6 climate scenarios [35,82].

While the limitations of vertical layering in the Bio-ORACLE v3.0 dataset focus on how the water-column is represented, an additional caveat concerns the horizontal resolution and habitat detail of benthic systems. Environmental data at an approximate 5 km resolution, while extremely highly resolved on a global scale, presents significant limitations, particularly

for benthic ecosystems where fine-scale habitat variability is crucial. One key issue is the lack of detailed global substrate information, as benthic species distributions are highly influenced by sediment type, rugosity, and microhabitat structures, which are often poorly resolved at coarse resolutions. This limitation can lead to oversimplification in habitat suitability models, potentially misrepresenting species distributions, especially in areas with heterogeneous seafloor characteristics. Higher-resolution data, when available, or local expert validation is therefore essential to improve accuracy, particularly for management and conservation applications. Lastly, Caspian Sea data were not available in BIO-ORACLE, hindering the AquaX framework to make projections in this region.

**Biogeographical range map limitation.** Another limitation of our methodology stems from the reliance on biogeographical provinces to approximate species' range maps when ERMs are unavailable. To achieve an ecologically meaningful representation, we derived distributions based on overlaps between valid occurrence records and established biogeographical divisions (PPOW and BPOW). We selected the province level for these divisions as a compromise between broader biome-level scales, which tend to overestimate distributions, and finer ecoregional scales, which are more sensitive to variations in sampling effort [83]. Nonetheless, the use of BR and PR might still overestimate actual species distributions. To mitigate this, we excluded provinces containing fewer than 3.3% of the total records for each species. Despite these precautions, inadequate sampling in high-sea or deep-sea environments can artificially truncate species ranges, potentially leading to underestimations. Such biases particularly affect deepwater benthic and highly mobile pelagic species as well as wider-ranging species [84,85]. Thus, expert validation remains essential to distinguish genuine ecological patterns from sampling artifacts.

It also needs to be highlighted that our exclusive focus on the marine environment excludes terrestrial and freshwater occurrences. Consequently, during our data cleaning process, occurrences recorded in freshwater or terrestrial habitats are systematically marked as unverified and omitted from the analysis. This may limit the accuracy of range predictions for species with significant estuarine or freshwater components in their life cycle (e.g., migratory anadromous fish species such as salmonids).

A further limitation is the model's restricted capacity to represent life-history traits of certain marine species, particularly in terms of invasive or non-native ranges. The AquaMaps bounding box delineates the "native range" of species, disregarding documented invasive occurrences unless explicitly shown in existing ERMs. For example, invasive species such as the red lionfish (*Pterois volitans,* Linnaeus, 1758) have well-documented distributions outside their native range (Schofield, 2010), yet these invasions are not captured unless pre-existing ERMs explicitly represent them (S11 Fig). This means our approach potentially underrepresents the full spatial extent and ecological impact of invasive marine species. In addition, our use of ocean basins to constrain species projections does not account for potential cross-basin dispersal pathways that may become increasingly relevant under climate change. For example, connections between the Indian and South Atlantic Oceans, or the Drake Passage linking the South Atlantic and South Pacific, represent open boundaries where species range shifts could occur. This limitation may lead to underestimation of new future areas for mobile or dispersive species capable of crossing such boundaries. Future work could address this by incorporating oceanographic connectivity through advection- or diffusion-based models, rather than relying solely on buffered range polygons.

**Implementation and testing of pseudo-absences.** Adding pseudo-absences (PAs) in species distribution models (SDMs) is a critical step that influences model accuracy and ecological validity [38,86]. Points where the species has not been recorded can be selected as either PAs or background data, depending on the specific modeling framework and underlying assumptions. Background data do not indicate where a species is present or absent [87]. Instead, background data remain consistent regardless of species observations, defining the environmental scope of the study. While background data describe the environmental conditions across the study area, PAs are specifically selected to improve model discrimination by approximating areas where the species is unlikely to occur [87].

One common approach is to select PAs from background data, meaning they are randomly distributed across the study area and can be placed in the same locations as presences. This method ensures that full environmental space is

considered, potentially capturing a wide range of ecological conditions [88]. However, randomly selecting PAs can introduce bias, especially if they overlap with areas that are environmentally suitable for the species, leading to over-prediction [89]. A more refined strategy is to place PAs outside the species' native range, which helps model range expansion under climate change or invasion scenarios [90]. This method prevents PAs from overlapping with presence areas but assumes that all extralimital areas are unsuitable, which may not be true in cases where species are dispersal-limited rather than environmentally constrained [91]. Lastly, selecting PAs within the species' native range by identifying areas of environmental dissimilarity can improve model performance by ensuring PAs reflect truly unsuitable conditions [92]. However, this method requires careful selection of environmental thresholds, as poorly chosen PAs may exclude viable habitats and reduce model generalizability [88,93]. Each of these strategies has trade-offs, and the optimal approach depends on the study objectives and species-specific ecological constraints.

In AquaX, we use the potential range (PR) of the species to place the PA, which includes a mix of suitable areas, potentially suitable areas and unsuitable areas for the species. Within this area, PAs are selected in environmentally unsuitable regions and strategically placed at the extremes of the species' niche to minimize the risk of assigning them to suitable habitats. This approach optimizes the study area to minimize computational costs while ensuring comprehensive coverage of the species' habitat, effectively capturing its full environmental niche (Fig 7). To generate PAs, we employed different strategies based on species ecology. For pelagic species, we used the Surface Range Envelope method, which is effective for organisms distributed in open water, where environmental conditions primarily dictate their presence. In contrast, for benthic and demersal species, we applied the disk method [39], a spatially explicit data partitioning option implemented in biomod2, where absences (and/or pseudo-absences) are excluded within a specified radius around presences. This method better accounts for their substrate-dependent distribution. These tailored approaches ensure that PAs align with species-specific environmental constraints, enhancing model accuracy and ecological relevance.

**Environmental niche model limitations.** ENM are simplified representations of reality, relying on assumptions about species-environment relationships based on observed occurrences. A key limitation is that ENMs predict habitat suitability indices (HSI) rather than actual species distributions, as they do not account for factors like dispersal, biotic interactions, or evolutionary adaptations. Uncertainties arise from incomplete knowledge of species' environmental tolerances and their ability to track shifting climatic conditions. Additionally, ENMs typically use coarse-resolution climate data (e.g., 50 km grids) which may not capture fine-scale habitat requirements. They also often neglect interactions between species, such as dependencies on prey or mutualists, which could indirectly influence species persistence under climate change. Furthermore, while climate change is a major driver, human activities such as bottom trawling, coastal development, and marine pollution also shape species distributions in ocean habitats but are not always integrated into projections [94–96]. The models also project species distribution changes driven by climate-induced environmental shifts. However, anthropogenic impacts, particularly habitat destruction, especially for reef-associated species, may lead to more severe biodiversity losses than those predicted solely by climate models. The exclusion of habitat loss from projections may result in an underestimation of real-world species range declines, as the destruction of critical habitats can impose a far greater risk to species survival than environmental shifts alone [97,98].

A final source of caution for users arises from differences in predictability between simpler statistical methods and more advanced machine learning (ML) algorithms. While contemporary ENM and SDM, particularly those utilizing ML algorithms, generally show superior performance based on common validation metrics such as Area Under Curve (AUC) and True Skill Statistic (TSS), these algorithms are inherently optimized to excel according to these very metrics [52,99]. However, a well-documented limitation of ML approaches is their tendency to overfit the training data, resulting in highly accurate representations of current distributions but potentially less reliable projections under changing environmental conditions, such as climate change [100,101]. In contrast, traditional algorithms, although often scoring lower on validation metrics for current distributions, typically assume a numerical representation closer to Hutchinson's ecological niche

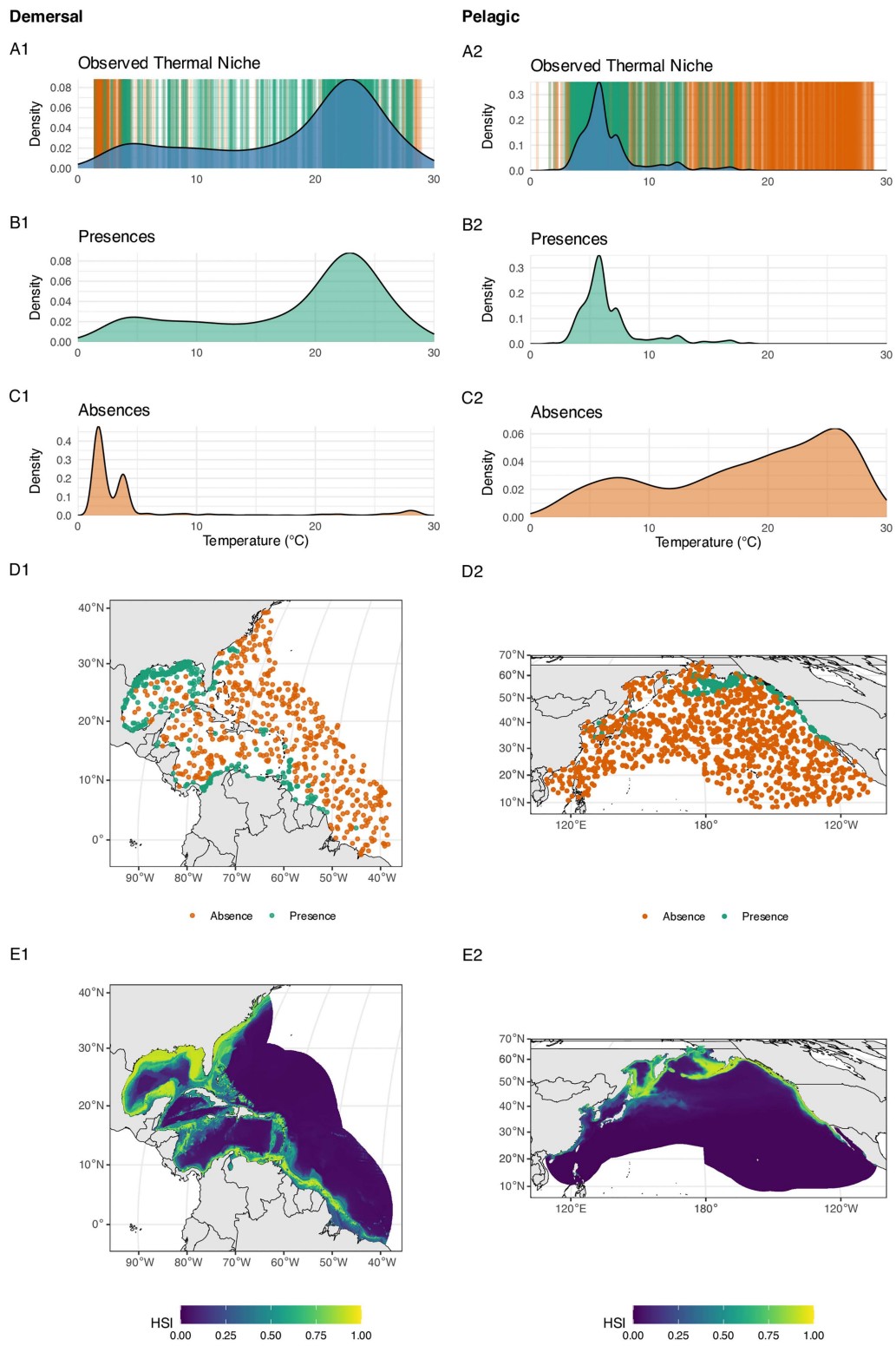

**Fig 7. Example of selection of pseudo-absences for demersal (first column) and pelagic (second column) species.** A) Observed thermal niche. Vertical lines show locations of presences (green) and pseudo-absences (orange); B) Density distribution of presences; C) density distribution of pseudo-absences; D) geographic distribution of occurrence and pseudo-absences; E) habitat suitability index for current (2000-2010) conditions. Made with Natural Earth. Free vector and raster map data @ naturalearthdata.com.

concept. This theoretical underpinning makes their predictions potentially more robust when extrapolating to novel or future environments [102]. Consequently, despite their lower immediate validation performance, classical algorithms may provide more realistic projections under future climate scenarios.

Due to these issues, we opted for computing an ensemble mean instead of a weighted ensemble mean, which would inherently favor ML-based models. This approach ensures a balanced representation, combining the strengths of both traditional and ML algorithms to generate ecologically meaningful and reliable projections for both current and future distributions.

## Conclusion

AquaX establishes a novel, comprehensive, and scalable framework for the delineation, modeling, and assessment of marine species distributions and biodiversity patterns. Building on the foundations of AquaMaps, AquaX integrates a series of methodological advances across multiple components: taxonomic standardization using WoRMS; rigorous occurrence data validation from AquaMaps, OBIS, and GBIF; ecological and biogeographical characterization through the D3OS classification; and the construction of expert- and data-informed range maps (BR and PR). It implements spatial and environmental thinning to minimize bias, introduces refined pseudo-absence generation tailored to species ecology, and employs ensemble ecological modeling based on ten distinct algorithms. High-resolution environmental data from Bio-ORACLE v3.0 allow species projections at spatial scales relevant to conservation decision-making, while validation metrics and spatial uncertainty maps enhance interpretability and transparency. Crucially, AquaX provides a structured interface between ERMs and SDMs, enabling iterative refinement and expert input. Designed as an open, modular, and evolving scientific resource, AquaX will continue to incorporate new data and methodological developments, supporting conservation planners, biodiversity scientists, and policymakers in the production of ecologically meaningful and policy-relevant marine species range assessments.

## Supporting information

**S1 Fig. Conceptual diagram of the process of compiling and standardizing AquaMaps, OBIS and GBIF occurrence data.** Abbreviations used: OCC – occurrence data, original_flags – column in OCC containing the original data-source flags (if available), GOODOCC – column in OCC containing standardized binary flags, basisOfRecord – column in OCC containing occurrence/observation type (e.g., human observation, machine observation, fossil specimen, museum collection).
(TIFF)

**S2 Fig. Diagram to represent species classification in the Distance-3 Ocean System (D3OS) system.**
(TIF)

**S3 Fig. Conceptual diagram of the process of creating biogeographic and potential ranges when an expert range map (ERM) is available and using it to flag occurrence data.** Abbreviations used: OCC – occurrence data, GOODOCC – column in OCC containing standardized binary flags, basisOfRecord – column in OCC containing occurrence type, BR – biogeographical range, PR – potential range, ERM – expert range map.
(TIFF)

**S4 Fig. Conceptual diagram of the process of creating biogeographic and potential ranges and using it to flag occurrence data when no expert range map is available.** Abbreviations used: OCC – occurrence data, GOODOCC – column in OCC containing standardized binary flags, BR – biogeographical range, PR – potential range, bbox – Aqua-Maps bounding box, Prov – Province, ERM – expert range map, N – number of observations in OCC.
(TIF)

**S5 Fig. Habitat Suitability Index (HSI) for *Thunnus obesus* (Lowe, 1839) across 10 modeling algorithms (average across the runs), with HSI values standardized to a 0–1 scale.** Made with Natural Earth. Free vector and raster map data @ naturalearthdata.com.
(TIFF)

**S6 Fig. Validation metrics for *Thunnus obesus* (Lowe, 1839) across all models.** Critical Score Index (CSI), Area Under the Curve (AUC), and True Skill Statistic (TSS) are reported for 300 model runs. Boxplots summarize the distribution of each metric for each model, with individual points representing outliers.
(TIFF)

**S7 Fig. UpSet plot showing missing value patterns across variables in all fish species (n = 18826) metadata.** The bar chart on the left indicates the total number of missing values for each variable. The matrix of connected dots in the center represents combinations of variables with simultaneous missing values, where a filled dot indicates missingness in that variable for a given subset of observations. The top bar chart quantifies the number of observations exhibiting each specific missing data pattern. Abbreviations: HAB- habitat type, DepthRangeDeep(Shallow)- max(min) depth of the species distribution range. Variable labels ending in "_NA" (e.g., "DepthRangeShallow_NA") indicate that the corresponding variable contains a missing (NA) entry; this notation is automatically assigned by the plotting function and does not refer to biological characteristics of the species.
(TIFF)

**S8 Fig. UpSet plot showing missing value patterns across variables in all molluscs (n = 51277) metadata.** The bar chart on the left indicates the total number of missing values for each variable. The matrix of connected dots in the center represents combinations of variables with simultaneous missing values, where a filled dot indicates missingness in that variable for a given subset of observations. The top bar chart quantifies the number of observations exhibiting each specific missing data pattern. Abbreviations: HAB- habitat type, DepthRangeDeep(Shallow)- max(min) depth of the species distribution range. Variable labels ending in "_NA" (e.g., "DepthRangeShallow_NA") indicate that the corresponding variable contains a missing (NA) entry; this notation is automatically assigned by the plotting function and does not refer to biological characteristics of the species.
(TIFF)

**S9 Fig. IUCN expert range map (yellow) and raw occurrence data of snake pipefish, *Hippocampus barbouri* (Jordan & Richardson, 1908).** Erroneous (flagged) occurrences are marked by red, verified (valid) occurrence points are marked in blue, records with no flags (unflagged) are marked as yellow. Depth ranges from BPOW are shaded in blues. Made with Natural Earth. Free vector and raster map data @ naturalearthdata.com.
(TIFF)

**S10 Fig. Distribution of snake pipefish, *Entelurus aequoreus* (Linnaeus, 1758).** A) Expert range map (shaded blue), Potential Range (shaded light violet) and raw occurrence data of snake pipefish. Erroneous (flagged) occurrences are marked by red, verified (valid) occurrence points are marked in blue, records with no flags (unflagged) are marked as yellow. B) Ensemble mean habitat suitability index (HSI) of snake pipefish. Orange line is the outline of the expert range map (IUCN ERM). Made with Natural Earth. Free vector and raster map data @ naturalearthdata.com.
(TIFF)

**S11 Fig. Biogeographical range (shaded light blue) and raw occurrence data of red lionfish, *Pterois volitans* (Linnaeus, 1758).** Erroneous (flagged) occurrences are marked by red, verified (valid) occurrence points are marked in blue, records with no flags are marked as yellow. Made with Natural Earth. Free vector and raster map data @ naturalearthdata.com.
(TIFF)

## Acknowledgments

FishBase and SeaLifeBase encoding teams for so many years of compiling, standardizing and entering data and information, with the help of successive librarians for the bibliographical references, in databases developed and maintained by the IT Group.

## Author contributions

**Conceptualization:** Gabriel Reygondeau, Derek P. Tittensor, Kristin Kaschner, William W. L. Cheung.

**Data curation:** Gabriel Reygondeau, Yulia Egorova, Kristin Kaschner, Kathleen Kesner-Reyes, Nicolas Bailly.

**Formal analysis:** Gabriel Reygondeau, Yulia Egorova, Kristina Boerder, Kristin Kaschner, Kathleen Kesner-Reyes, Nicolas Bailly, William W. L. Cheung.

**Funding acquisition:** Gabriel Reygondeau, Kristina Boerder, Derek P. Tittensor, William W. L. Cheung.

**Investigation:** Gabriel Reygondeau, Yulia Egorova, Kristin Kaschner, Nicolas Bailly.

**Methodology:** Gabriel Reygondeau, Yulia Egorova, Derek P. Tittensor, Kristin Kaschner, Kathleen Kesner-Reyes, Nicolas Bailly, William W. L. Cheung.

**Project administration:** Gabriel Reygondeau, Kristina Boerder, William W. L. Cheung.

**Resources:** Gabriel Reygondeau, William W. L. Cheung.

**Software:** Gabriel Reygondeau, Yulia Egorova.

**Supervision:** Gabriel Reygondeau, William W. L. Cheung.

**Validation:** Gabriel Reygondeau, Yulia Egorova, Derek P. Tittensor.

**Visualization:** Gabriel Reygondeau, Yulia Egorova, Kristina Boerder, Derek P. Tittensor, Kristin Kaschner, Kathleen Kesner-Reyes, Nicolas Bailly, William W. L. Cheung.

**Writing – original draft:** Gabriel Reygondeau, Yulia Egorova, Kristina Boerder, Derek P. Tittensor, Kristin Kaschner, Kathleen Kesner-Reyes, Nicolas Bailly, William W. L. Cheung.

**Writing – review & editing:** Gabriel Reygondeau, Yulia Egorova, Kristina Boerder, Derek P. Tittensor, Kristin Kaschner, Kathleen Kesner-Reyes, Nicolas Bailly, William W. L. Cheung.

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
