## [Decision Letter · Decision Letter 0]

10 Nov 2025

Dear Dr. Reygondeau,

Thank you for submitting your manuscript to PLOS ONE. After careful consideration, we feel that it has merit but does not fully meet PLOS ONE’s publication criteria as it currently stands. Therefore, we invite you to submit a revised version of the manuscript that addresses the minor points raised during the review process.

We look forward to receiving your revised manuscript.

Kind regards,

Athanassios C. Tsikliras

Academic Editor

PLOS ONE

Journal Requirements:

3. Thank you for uploading your study's underlying data set. Unfortunately, the repository you have noted in your Data Availability statement does not qualify as an acceptable data repository according to PLOS's standards.

4. We note that Figure 2, 3, 4, 5, 6, 7, S9, S10, and S11 in your submission contain map images which may be copyrighted. All PLOS content is published under the Creative Commons Attribution License (CC BY 4.0), which means that the manuscript, images, and Supporting Information files will be freely available online, and any third party is permitted to access, download, copy, distribute, and use these materials in any way, even commercially, with proper attribution. For these reasons, we cannot publish previously copyrighted maps or satellite images created using proprietary data, such as Google software (Google Maps, Street View, and Earth). For more information, see our copyright guidelines: http://journals.plos.org/plosone/s/licenses-and-copyright.

1. You may seek permission from the original copyright holder of Figure 2, 3, 4, 5, 6, 7, S9, S10 and S11 to publish the content specifically under the CC BY 4.0 license.

5. Please remove your figures from within your manuscript file, leaving only the individual TIFF/EPS image files, uploaded separately. These will be automatically included in the reviewers’ PDF.

6. We are unable to open your Supporting Information file “manuscript_script.Rmd.zip” Please kindly revise as necessary and re-upload.

7. We notice that your supplementary figures are included in the manuscript file. Please remove them and upload them with the file type 'Supporting Information'. Please ensure that each Supporting Information file has a legend listed in the manuscript after the references list.

Reviewer's Responses to Questions

**Comments to the Author**

1. Is the manuscript technically sound, and do the data support the conclusions?

Reviewer #1: Yes

Reviewer #2: Yes

2. Has the statistical analysis been performed appropriately and rigorously?

Reviewer #1: Yes

Reviewer #2: I Don't Know

3. Have the authors made all data underlying the findings in their manuscript fully available?

Reviewer #1: Yes

Reviewer #2: Yes

4. Is the manuscript presented in an intelligible fashion and written in standard English?

Reviewer #1: Yes

Reviewer #2: Yes

Reviewer #1: This paper describes the next-generation version of the AquaMaps Ecological Niche Model, named AquaX, intended as an operational methodology that can, in theory, be adapted to many species and environmental data after addressing the challenges associated with integrating multi-source data.

Due to its combination of simplicity and effectiveness, AquaMaps has served as a reference and the primary guide for approximately assessing the presence of many species, including rare ones. I recognise that proposing a new alternative model capable of producing detailed maps for over 30,000 species is ambitious; however, it holds significant potential for global conservation and planning. Therefore, this paper has great potential to become a reference in the field.

Generally, the paper is sound and comprehensive. The Introduction effectively explains the context and objectives of the study. The procedure for assessing occurrence quality greatly extends that of the previous model. The innovative management of ecological and biogeographical information is both reasonable and intriguing. The implementation of range maps is also interesting and replicable. I especially appreciated the multi-scale approach and differentiation by depth layer.

Overall, the methodology exemplifies an integrative system that leverages publicly available knowledge and repositories, along with a novel modelling approach well distinct from AquaMaps. The paper reflects the authors' extensive effort, with a thorough discussion and a results section that explores various applications of the methodology.

Specific Comments:

- Lines 194-195: Is the GOODOCC set to 1 in all other cases?

- Line 201: Is NA equivalent to the "blank" value you assigned for OBIS? Could you standardize the notation, please?

- Lines 206-210: I do not understand how GBIF is utilized in this process.

- Lines 300-304: How are blank and NA occurrences handled in this process?

- Lines 415-418: This appears to be traditional clustering. How did you identify the distinct environmental combinations?

- Line 445: Please remove the initial "(".

- Lines 445-453: Please specify why you selected these models over others or a subset. Many methods overlap in their foundational assumptions, particularly in the partitioning of the feature space during modelling (e.g., tree-based algorithms). Only a few methods, such as neural networks, can model complex relationships between data. It is possible that different models may exhibit very similar behaviour, leading the ensemble assessments to reflect less complex inter-data relationships.

- Line 628: Shouldn't it be "(Ginsburg, 1930)", with parentheses?

- Line 692: "Information" should be written in lowercase.

Reviewer #2: The study by Reygondeau et al. titled “AquaX: An enhanced and revised AquaMaps framework to model marine species distributions and biodiversity” presents the upgraded version of AquaMaps, a valuable tool that models and provides global maps of current and future/potential species distributions. The manuscript is generally very well written and thoroughly describes the methodology using high quality, informative figures. I particularly appreciate the extensive, detailed, and honest section on the multi-leveled limitations of the method. Despite those, AquaX remains a valuable tool that is expected to be extensively used globally. I suggest minor revisions and provide specific comments below. One important comment has to do with polygon buffering. Other comments focus on enhancing certain discussion parts with relevant references. Because the results are (understandably) presented together with the discussion, sometimes the discussion portion lacks broader context through citations.

**Do you want your identity to be public for this peer review?** For information about this choice, including consent withdrawal, please see our Privacy Policy

Reviewer #1: No

Reviewer #2: No

---

## [Author Response · Author response to Decision Letter 1]

8 Dec 2025

Editors’ Comments

Answer: The content of the revised manuscript and file names has been adjusted to meet the journal style requirements

Answer: The ORCID iD is added to the corresponding author account.

3. Thank you for uploading your study's underlying data set. Unfortunately, the repository you have noted in your Data Availability statement does not qualify as an acceptable data repository according to PLOS's standards.

Answer: Manuscript data has been deposit to Figshare DOI: 10.6084/m9.figshare.30754358

4. We note that Figure 2, 3, 4, 5, 6, 7, S9, S10, and S11 in your submission contain map images which may be copyrighted. All PLOS content is published under the Creative Commons Attribution License (CC BY 4.0), which means that the manuscript, images, and Supporting Information files will be freely available online, and any third party is permitted to access, download, copy, distribute, and use these materials in any way, even commercially, with proper attribution. For these reasons, we cannot publish previously copyrighted maps or satellite images created using proprietary data, such as Google software (Google Maps, Street View, and Earth). For more information, see our copyright guidelines: http://journals.plos.org/plosone/s/licenses-and-copyright.

1). You may seek permission from the original copyright holder of Figure 2, 3, 4, 5, 6, 7, S9, S10 and S11 to publish the content specifically under the CC BY 4.0 license.

2).If you are unable to obtain permission from the original copyright holder to publish these figures under the CC BY 4.0 license or if the copyright holder’s requirements are incompatible with the CC BY 4.0 license, please either i) remove the figure or ii) supply a replacement figure that complies with the CC BY 4.0 license. Please check copyright information on all replacement figures and update the figure caption with source information. If applicable, please specify in the figure caption text when a figure is similar but not identical to the original image and is therefore for illustrative purposes only.

Answer: all maps were produced using tmap package that utilized natural Earth data (public domain). Each figure caption that show map now contains the following attribution: Map data were obtained from Natural Earth (public domain; naturalearthdata.com). Attributions were also given to any images that use animal silhouettes.

5. Please remove your figures from within your manuscript file, leaving only the individual TIFF/EPS image files, uploaded separately. These will be automatically included in the reviewers’ PDF.

Answer: All figures have been removed from the manuscript file. Only the figure captions remain in the manuscript, placed in the correct locations according to PLOS guidelines. All corresponding TIFF image files have been uploaded separately as individual figure files. We confirm that no figures are embedded in the manuscript, and the submission now conforms fully to the PLOS figure-upload requirements.

6. We are unable to open your Supporting Information file “manuscript_script.Rmd.zip” Please kindly revise as necessary and re-upload.

Answer: manuscript_script.Rmd.zip was autogenerated during submission process and us erroneous. he full analysis script (manuscript_script.Rmd) used to generate the results and figures has been deposited on Figshare together with the dataset and is accessible at https://doi.org/10.6084/m9.figshare.30754358

7. We notice that your supplementary figures are included in the manuscript file. Please remove them and upload them with the file type 'Supporting Information'. Please ensure that each Supporting Information file has a legend listed in the manuscript after the references list.

Answer: We notice that your supplementary figures are included in the manuscript file. Please remove them and upload them with the file type 'Supporting Information'. Please ensure that each Supporting Information file has a legend listed in the manuscript after the references list.

Answer: we have followed reviewer suggestion to add more references to support our claims

Answer: We have thoroughly reviewed our reference list for completeness, accuracy, and retracted publications. During this process, we (i) removed references that are no longer cited in the revised manuscript, (ii) updated and corrected citation details where needed, and (iii) added several recent and relevant studies to better reflect the current state of the field. We have replaced the original reference list with a new, streamlined list in the revised manuscript, which now complies with the PLOS ONE reference and citation style. We checked all cited journal articles and reports against the publisher websites and major bibliographic databases and, to the best of our knowledge, none of the references currently cited in the manuscript have been retracted. Accordingly, no retraction notices needed to be added and no rationale for citing retracted work was required.

Reviewer #1:

This paper describes the next-generation version of the AquaMaps Ecological Niche Model, named AquaX, intended as an operational methodology that can, in theory, be adapted to many species and environmental data after addressing the challenges associated with integrating multi-source data.

Due to its combination of simplicity and effectiveness, AquaMaps has served as a reference and the primary guide for approximately assessing the presence of many species, including rare ones. I recognize that proposing a new alternative model capable of producing detailed maps for over 30,000 species is ambitious; however, it holds significant potential for global conservation and planning. Therefore, this paper has great potential to become a reference in the field.

Generally, the paper is sound and comprehensive. The Introduction effectively explains the context and objectives of the study. The procedure for assessing occurrence quality greatly extends that of the previous model. The innovative management of ecological and biogeographical information is both reasonable and intriguing. The implementation of range maps is also interesting and replicable. I especially appreciated the multi-scale approach and differentiation by depth layer.

Overall, the methodology exemplifies an integrative system that leverages publicly available knowledge and repositories, along with a novel modelling approach well distinct from AquaMaps. The paper reflects the authors' extensive effort, with a thorough discussion and a results section that explores various applications of the methodology.

Answer: we would like to thank the reviewer for positive feedback and for valuable comments thought the manuscript

Specific Comments:

+ Lines 194-195: Is the GOODOCC set to 1 in all other cases?

Answer: in this sentence we were talking about OBIS original flags ( as shown in diagram S1_Fig1)

There is no single standardized set of flags as different databases flag tehir datat differently. That is why we have 2 sets of flag columns: 1) the original data-source flags (original_flags) and 2) the standardized binary flags (GOODOCC). The content of original_flags depends on the provider. For example, GBIF does not supply native flags, so this column remains empty, as shown in the diagram. In contrast, OBIS provides extensive data-quality flags (see OBIS Manual: “18 Data quality flags”, https://manual.obis.org/dataquality.html), and these are recorded in original_flags. As indicated in the diagram, occurrences are assigned a standardized flag value of 0 only when the original OBIS flags indicate that the point falls on land, has coordinates of (0,0), or contains other latitude/longitude errors. The rest of the flags are ignored (not flagged), and their verifications is performed later in the process (S1 Figs 3-4). AquaMaps uses a binary flagging system in which 1 indicates a good occurrence and 0 indicates a bad occurrence, as described in the manuscript. For this reason, all flags are standardized to a binary format and stored in the GOODOCC column. The data-cleaning steps (ie further assigning flags for occurrence points that do not have flags) then continue as outlined in subsequent diagrams.

We re-wrote the sentence in the following way: Other original OBIS flags, such as missing depth information or occurrences falling outside known bathymetric limits, were retained in original flag column but not automatically considered erroneous, and their GOODOCC value was left blank in standardized binary flag column (GOODOCC).

+ Line 201: Is NA equivalent to the "blank" value you assigned for OBIS? Could you standardize the notation, please?

Answer: thank you for pointing out this inconsistence. The sentence is re-written as following: “Since GBIF does not include explicit quality flags in its dataset, and only records without spatial issues and with valid coordinates were downloaded, the flag column for all GBIF entries was left blank to reflect the absence of a source-provided quality classification.”

+ Lines 206-210: I do not understand how GBIF is utilized in this process.

Answer: to illustrate the process : below you can find the hypothetical example of the duplicated data: we consider this entries duplicated because the coordinates and time of collection is identical in this case. One records come from AquaMaps database, the other comes form GBIF. The only difference here that this record was validated by AquaMaps team and flagged as good ( GOODOCC column =1) while GBIF has unflagged data (NA in GOODOCC). Hence in order not to store the duplicated information, we decided to keep the most complete one: in this case is the occurrence record from AquaMaps, as GBIF record does not contribute any extra information here.

latitude longitude dd/mm/yyyy DataBase GOODOCC AphiaID

10 10 10/10/2010 AquaMaps 1 111111

10 10 10/10/2010 GBIF NA 111111

+ Lines 300-304: How are blank and NA occurrences handled in this process?

Answer: We remove only records explicitly flagged as bad (GODOCC = 0). When the flag is blank, we cannot determine whether the record is good or bad; therefore, for defining the bounding box we retain all available records, including those flagged as 1 or left blank.

+ Lines 415-418: This appears to be traditional clustering. How did you identify the distinct environmental combinations?

Answer: To identify distinct environmental combinations, each environmental variable was first rounded to predefined, ecologically meaningful increments (Table 2). Rounding converts continuous variables into discrete bins; each bin corresponds to a single category for that variable. Unique environments were then defined as every unique combination of these binned values across all variables. This produces a set of discrete, multidimensional environmental categories rather than clusters generated by an algorithm. Each occurrence record was assigned to one such category, and one record was randomly retained per category to ensure balanced environmental representation.

+ Line 445: Please remove the initial "(".

Answer: The issue has been addressed in accordance with the reviewer’s suggestion.

+ Lines 445-453: Please specify why you selected these models over others or a subset. Many methods overlap in their foundational assumptions, particularly in the partitioning of the feature space during modelling (e.g., tree-based algorithms). Only a few methods, such as neural networks, can model complex relationships between data. It is possible that different models may exhibit very similar behaviour, leading the ensemble assessments to reflect less complex inter-data relationships.

Answer: the matter is discussed in Results and discussion section (final major update in the subsection on the evolution of AquaMaps methodology):

“The final major update of AquaX employs an ensemble modeling approach comprising ten independent models, inspired by methodologies utilized in the FISH-MIP and CMIP projects [56,57]. Each model was specifically selected to encompass various types of ENM and species distribution modeling (SDM) techniques, including linear models (GLM, GAM, MARS), background or entropy-based models (MAXNET), explorative stat

---

## [Decision Letter · Decision Letter 1]

10 Dec 2025

AquaX: An enhanced and revised AquaMaps framework to model marine species distributions and biodiversity

PONE-D-25-56229R1

Dear Dr. Reygondeau,

We’re pleased to inform you that your manuscript has been judged scientifically suitable for publication and will be formally accepted for publication once it meets all outstanding technical requirements.

Kind regards,

Athanassios C. Tsikliras

Academic Editor

PLOS One

Additional Editor Comments (optional):

Reviewers' comments:

Reviewer's Responses to Questions

**Comments to the Author**

Reviewer #1: All comments have been addressed

2. Is the manuscript technically sound, and do the data support the conclusions?

Reviewer #1: Yes

3. Has the statistical analysis been performed appropriately and rigorously?

Reviewer #1: Yes

4. Have the authors made all data underlying the findings in their manuscript fully available?

Reviewer #1: Yes

5. Is the manuscript presented in an intelligible fashion and written in standard English?

Reviewer #1: Yes

Reviewer #1: All comments have been addressed. The paper is now ready for publication, from my point of view. I think that the novelty and the methodology would deserve an appropriate highlighting on the journal's website.

**Do you want your identity to be public for this peer review?** For information about this choice, including consent withdrawal, please see our Privacy Policy

Reviewer #1: No

---

## [Editor Report · Acceptance letter]

PONE-D-25-56229R1

PLOS One

Dear Dr. Reygondeau,

I'm pleased to inform you that your manuscript has been deemed suitable for publication in PLOS One. Congratulations! Your manuscript is now being handed over to our production team.

Kind regards,

on behalf of

Professor Athanassios C. Tsikliras

Academic Editor

PLOS One